# Societies in balance: Monumentality and feasting activities among southern Naga communities, Northeast India

Maria Wunderlich[1]*, Tiatoshi Jamir[2], Johannes Müller[1], Knut Rassman[3], Ditamulü Vasa[2]

1 Institute for Pre- and Protohistoric Archaeology, Christian-Albrechts-University, Kiel, Germany,
2 Department of History & Archaeology, Nagaland University, Kohima, India, 3 Romano-Germanic Commission, German Archaeological Institute, Frankfurt a. M., Germany

* m.wunderlich@ufg.uni-kiel.de

**Data Availability Statement:** The underlying data of the box plots and graphs are available within the data repository (Zenodo.org) under the following

## Abstract

Among various Naga communities of Northeast India, megalithic building and feasting activities played an integral role in the different and intertwined dimensions of social and political organisation until very recently. During a collaborative fieldwork in 2016, we visited different village communities in the southern areas of Nagaland and recorded local knowledge about the function and social implications of megalithic building activities. The preserved knowledge of the monuments themselves and their embeddedness in complex feasting activities and social structures illustrate the multifaceted character of megalithic building. The case study of Nagaland highlights how the construction of megalithic monuments may fulfil very different functions in societies characterised by institutionalised hierarchies than in those that have a more egalitarian social organisation. The case study of southern Naga communities not only shows the importance of various dimensions and courses of action–such as sharing and cooperation, competitive behaviour, and the influence of economic inequality–, but also the importance of social networks and different layers of kinship. The multifaceted and interwoven character of megalithic building activities in this ethnoarchaeological case study constitutes an expansion for the interpretation of archaeological case studies of monumentality.

## Introduction

Monumentality represents one of the most enduring and diversified topics within archaeological research. Due to their impressive size and visibility, monuments have been an object of archaeological research from the earliest phases of the discipline onwards. Within the ever-changing course of interpretative frameworks connected to monumentality, including megalithic monuments, archaeologists have focused on their form and typology, their position within the socially constructed landscape, and their social meaning. Although comparisons with recent examples of megalithic construction have been made (e.g. [1–3]), explicitly comparative studies on the social implications of this specific phenomenon are still rare.

DOI: 10.5281/zenodo.4494265 (http://doi.org/10.5281/zenodo.4494265).

**Funding:** This research was funded by German Research Foundation (DFG) within the Priority Program "Early Monumentality and Social Differentiation" (SPP 1400), subproject "Equality and Inequality" (MU 1259/18-3, project no. 238040975; PI: JM). The finalization of results took place in the frame of the Collaborative Research Centre SFB 1266 "Scales of Transformations" (Deutsche Forschungsgemeinschaft (DFG) – project number 290391021) and the Cluster of Excellence EXC 2150 "ROOTS - Social, Environmental, and Cultural Connectivity in Past Societies" (Deutsche Forschungsgemeinschaft (DFG) – project number 390870439). The funders had no role in study design, data collection and analysis, decision to publish, or preparation of the manuscript.

**Competing interests:** The authors have declared that no competing interests exist.

Our research in the Indian state of Nagaland follows such a comparative approach [3–5]. We documented the entanglement of various factors relating to recent megalithic building activities, such as political organisation, kinship-systems, ritual behavior, and economic aspects. The resulting dataset allows an in-depth analysis of the recursive relationship between megalithic structures and the different layers of communities building them. The analyses of the different types of megalithic monuments, their affiliation with social groups, and their placement within the structured landscape are exemplified here with one specific village of the Chakhesang Naga, the village of Rünguzu, which was among the locations of our fieldwork in 2016. Since megalithic building activities among Naga communities cannot be understood without a consideration of the accompanying feasting activities, we also present an outline of this practice. The results derived from this dataset allow the identification of the underlying social structures and mechanisms, which can, in turn, be used to enhance our understanding of archaeological case studies.

Northeast India is a region with an extraordinarily high density of and variability in recent megalithic building activities, with a diversity and depth that are not sufficiently recognised within European megalithic research. In Northeast India in general, and in Nagaland in particular, megalithic building took place in different places, at different times, and within varying contexts of social organisation. Megalith building traditions were documented in the early 20[th] century by colonial officials of the English colonial government among different Naga groups [6–15]. The social organisation of these groups was described as ranging from chiefdom-like structures (among the Konyak Naga) to non-institutionalised, flat hierarchies (among the Angami Naga). In these contexts, megalithic building developed in very different ways, representing and materialising fundamentally different social mechanisms and influences ([16], 110f., [17–20]). Within the past century, megalith building and many of the associated social institutions and traditions have disappeared due to the fundamental changes that took place following the introduction of Christianity and the integration of Nagaland into the national state of India (cf. [21]). However, local knowledge about past social institutions and the associated structures, and also the function of megalithic building, is in many cases still preserved.

The biggest potential for ethnoarchaeological research lies in its suitability to present in-depth analysis of specific phenomena and their societal entanglement. Deriving from this archaeological and anthropological background, three main questions may be posed:

First, which social relations are meaningful and materialised within monumental constructions and how is the recursive relationship between relatedness and monuments shaped and maintained?

Second, where can we place megalithic building activities within the range of collective and individualised, as well as communal and exclusionary strategies?

Third, how are monuments entangled within the processes of landscape construction of the communities building them?

The case study of Nagaland presents an example of the construction, perception, and maintenance of kinship structures and their connection to monumentality. The case study shows how deeply rooted and diverse megalithic building is within the different social groups, with partly very different courses of action.

## Megalithic monuments in Northeast India

The remains of extensive megalithic building activities can be found all over the different regions of Northeast India and span different contexts as well as types (Fig 1). Particularly well-known are the monuments built by the Khasis, the Naga, the Garo, the Karbis, the Tiwas, and the Kuki-Chin-Mizo-groups. These communities are found dispersed in the states of

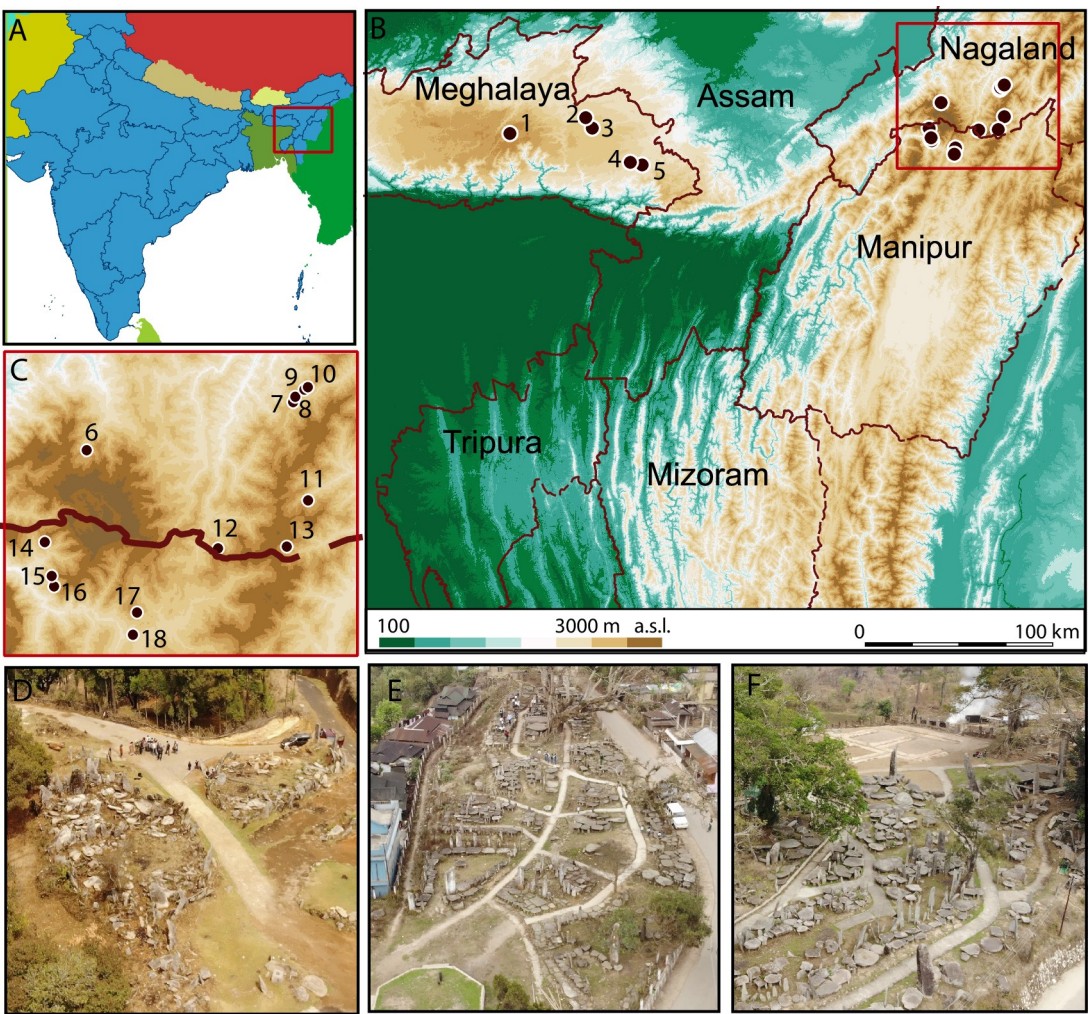

**Fig 1. Overview of the research area in Northeast India.** A-C: sites with field surveys, D: Myllem, Meghalaya, E: Nartiang, Meghalaya, F: Nongbag, Meghalaya (Data provisioning and graphic: RGK Frankfurt, UFG Kiel; shape files made with Natural Earth, DEM based on CGIAR-CSI SRTM data. DEM republished from Jarvis A., H.I. Reuter, A. Nelson, E. Guevara, 2008, Hole-filled seamless SRTM data V4, International Centre for Tropical Agriculture (CIAT), available from http://srtm.csi.cgiar.org, under a CC BY license, with permission from Alliance of Bioversity International and CIAT, original copyright 2004–2021). 1 Myllem, Meghalaya, 2 Nartiang, Meghalaya, 3 Nangbah, Meghalaya, 4 Mookynbah, Meghalaya, 5 Tongseng, Meghalaya, 6 Khonoma, Nagaland, 7 Chozuba, Nagaland, 8 Rünguzu, Nagaland, 9 Yoruba, Nagaland, 10 Rhüzazho, Nagaland, 11 Khezhakeno, Nagaland, 12 Zhavame, Nagaland, 13 Ze Mnui, Manipur, 14 Willong, Manipur, Sekume, 15 Maram, Manipur, 16 Maram-Sagonbam, Manipur.

Assam, Meghalaya, Nagaland, and Manipur, although especially Naga communities are also found living in the adjoining regions of Myanmar. The recent edited volume by Marak [22] is the first publication to present a comprehensive regional overview of the diverse megalithic practices in Northeast India, and it does so from a wide array of perspectives, ranging from typo-morphological variation, dating of megaliths, monumentality and complex village polities, the socio-economic dynamics and their interplay with ideas of monumentality, feasting and traditional architecture, landscape and social memory, mortuary behaviour, megaliths as vital communicative forms and substance of social life, to issues on notions of ideology.

The menhir type, in variety of sizes, seems to be the most common type throughout Northeast India. Menhirs either commemorate social events or memorialise the dead (e.g. Naga,

Khasi-Jaintia, Mizo, Karbi), while dolmens are either mostly associated with the ancestress of the clan (among the Khasi-Jaintia, [23]) or are raised above burials as memorials (among Zemi Naga [24] and Manipur Naga [25]). Cist burials, although they differ in form and meaning, are built either as ossuary interment facilities for the family/clan (among the Khasi-Jaintias, [26, 27]) or as primary inhumation structures for the deceased members of the community (among the Angami and Chakhesang Naga, [17, 18, 28, 29]).

A common characteristic shared among all of these groups is the fact that megalithic building is a recent phenomenon and that it was almost entirely abandoned within the last decades, since about 1960, due to the rise of Christianity (cf. [30]). However, megalithic building practices have somehow survived in small pockets amongst some communities, such as the Naga of Manipur [25], the Karbis of Karbi Anglong District and the Dimoria area of the Kamrup District of Assam [31, 32], and the Jaintias of Meghalaya [33], who have retained their traditional religion. The megalithic monuments represented in these communities are variable and diverse in type, and hence only a brief overview is given here. The megaliths in Meghalaya are unusual in that they are strongly connected to the representation of matriliny ([34], 73f.).

Although many aspects of these traditions, such as their age, temporality, and origin (cf. [5], 18, [35]), will require further investigation, an extensive body of published literature and theses from the past few years shed some light. In the case of the Khasis, although the top layer of the site of Myrkhan in the Khasi-Jaintia-Hills has yet to be dated, the excavators tentatively place the beginning of the megalithic culture in the Khasi-Jaintia Hills in the 1st century BC to 1st century AD ([36], [37], 15). Two additional significant [14]C dates documenting the potential start of these practices come from the Kachari megalithic ruins at Rajbari (Dimapur, Nagaland), from an excavation undertaken by Nienu [38] provides two [14]C dates for the Rajbari site: 1530±180 (AD 270–660) and 1300±180 (AD 570–940) ([39], 212–240). A few radiocarbon dates are now available for new excavations in Nagaland. The site of Chungliyimti associated with the origin myth of six stones, or *Longtrok*, is dated to 910±70 BP and 1020±80 BP [40, 41], while the sites of Khezhakeno, Movolomi, Khusomi, and Phor, all found in association with stone monuments, are dated to 500±50 BP (cal AD 1320–1350), 410±60 BP (cal AD 1420–1640), 530±40 BP (cal AD 1320–1350) and 230±60 BP (cal AD 1500–1600) [42].

While in some societies that exhibit recent megalithic building traditions these are linked to mortuary practices (e.g. in the case of Khasi and Jaintia; [23]), this is by no means always the case and these traditions are subject to great variability and time-depth.

## Standing stones

Standing stones of various sub-types form the backbone and most frequent occurrence of megalithic monumentality in Northeast India. The standing stones, in stark contrast to many of the dolmens and cairns, are not directly linked to mortuary rites (cf. [43], [44], 38). Standing stones are usually linked to complex feasting activities and serve either as commemorative monuments representing the prestige and social position of the monument builder, or as memorials for the deceased builder or his relatives ([16], 110, [45], 650). The size of the menhirs, as well as the number of stones per monument, varies considerably among all societies engaged in megalithic construction in parts of Northeast India. They may reach heights of up to 5 metres or be as small as 60 centimetres (Fig 2; compare [45], 649).

Furthermore, a wide range of variation exists with regard to the arrangement and number of stones. Standing stone monuments may consist of only one standing stone, or they may consist of many stones, which may form an alignment, or avenue. There is a great amount of regional and local variability connected to these factors. Further, standing stones may be erected on a platform formed of smaller stones or bricks or they may be surrounded by a

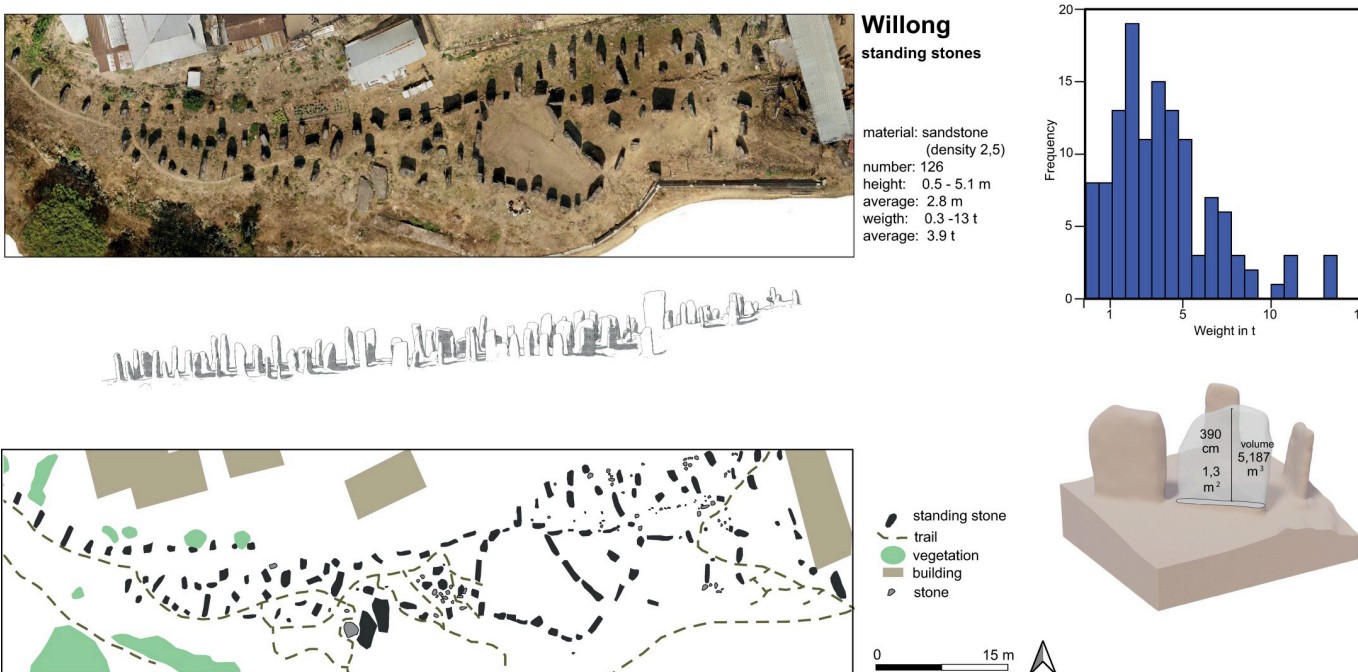

**Fig 2. Willong, Manipur.** Structure from Motion (SfM) based map and quantitative data (volume, weight) on the standing stones (Data provisioning and graphic: RGK Frankfurt).

frame of stones (cf. [3], 191). Despite the individual translations of this type of megalithic building, the shared framework in most cases is the interconnectedness of the stones with feasting practices and social prestige.

In a minority of cases, and especially among the Mizos in Mizoram, standing stones may be elaborately decorated with carvings. These mainly depict humans as warriors, as well as material items, such as gongs, but they also show animal depictions ([46], 128, [47]).

## Sitting platforms (*tehuba*)

Sitting platforms (Angami: *tehuba*) constitute the second main type of megalithic monumentality in Northeast India and, similar to the various types of standing stones, are found among many of the different groups exhibiting megalithic building traditions.

The form and, more importantly, the association of these monuments is quite diverse. Within Angami Naga communities, sitting platforms may be constructed by individual feast-givers, or by a collective group, such as a *khel* or clan ([3], 199–207). Furthermore, examples of sitting platforms that have been raised over the graves of feast-givers are present among the Angami Naga in southern Nagaland ([18], 629).

## Dolmens and cairns

The frequency in which dolmens are found varies greatly and depends on the specific social context. Although dolmens are not found among the Angami Naga and the Chakhesang Naga, they are present among the Poumai, the Maram, the Mao and the Liangmai in Manipur ([45], 655, [48], 189), as well as in Meghalaya [23, 49] and Mizoram ([46], 133). Among the Poumai, dolmens signify a high social standing of the monument builder and fulfil a very similar function and role to the standing stones among the Angami Naga and Chakhesang Naga, while

cairns may also serve as burial monuments ([50], 114). Dolmens of various types were frequently erected among the Khasi in Meghalaya. These monuments may be subdivided into different types, fulfilling different purposes. This dolmen type includes table stones that serve as a resting place, as well as table stones with accompanying standing stones that serve as a market place. However, burial cists are the most important among this dolmen type. Although table stones are commemorative markers of the completion of funerary rites and diverse rituals, the cists are directly used for funerary practices and the placement of cremated bones ([51] 167ff.).

Of special interest in relation to cairns are recent excavations of burials in the Angami region. In combination with interviews undertaken with village elders, these revealed manifold distinctions among burial rites, comprising burials outside and inside the village, as well as grave construction with or without stone architecture (cf. [18], 622–25). Within this system, social roles, but also the circumstances of death, played a distinctive and determining role, displayed by means of the grave items and the form and structure of the grave. The use of capstones and stone structures could be part of the burial rites and was, similarly to the standing stones, a display of wealth and status of the deceased (cf. [18], 625).

### Stones connected to specific actions or social roles

Although the previously described monument types of standing stones and sitting platforms, as well as dolmens and cairns, comprise the vast majority of stone monuments in Northeast India, other types of monuments do exist. These monuments, which are erected with reference to specific actions and social roles, span a wide range of forms, including specific stones for head-hunters (e.g. [50], 116) or stones representing the affairs of men, thereby displaying their libido prowess (e.g. [16], 110).

## A comparative approach using bottom-up perspectives

### The archaeology of megalithic monuments and bottom-up perspectives

Within archaeological research, megalithic monuments as a dimension of the broad topic of monumentality have always been an extensively researched and analysed topic (cf. [52]). Across different archaeological phases and very different socio-economic contexts, megalithic monuments are a recurring feature, thus providing a suitable, yet strikingly variable, phenomenon for comparative research questions. The range of case studies is as wide as the range of different interpretations of these monuments. Among the most influential are certainly their interpretation as territorial markers [53, 54]; as symbols for the desire of humans to domesticate nature [55]; and as a materialisation of a specific, group-related identity (e.g. [56, 57]). The different approaches can roughly be summarised within a more functional, symbolical, ideological, or social interpretative background.

Despite the range of interpretation connected to megalithic monuments, researchers have often perceived them from an exclusionary perspective. The research focus was repeatedly put on mechanisms potentially excluding wider parts of communities from using the monuments. This perspective incorporates basic assumptions about a need of centralised labour organisation, as well as the use of these monuments with the help of established leadership-structures or by specific groups of individuals within the framework of exclusionary strategies. This includes interpretations of the construction of megalithic monuments as a representation of specific social groups seeking influence, or as symbols of power (e.g. [1, 58–60]). Although these interpretations will certainly be appropriate for some case studies, there are also a number of alternate interpretations of megalithic monuments. A valid counterpoint to the perception of megaliths as a materialisation of exclusionary strategies can be drawn from the examination of recent case studies. In this regard, bottom-up perspectives seek to place the focus on individual

agency and the underlying social mechanisms that influence interaction and societal organisation (e.g. [61]; with a focus on the organisation of agricultural activities, [62]).

Megalithic monuments potentially exhibit a broad range of manifestations, since they are very often an accumulation of starkly different forms of human behaviour, social mechanisms and social relations (cf. [63]). Therefore, they are particularly suitable for the application of bottom-up perspectives, focusing on underlying choices and mechanisms carried out by individuals belonging to a specific community. This kind of perspective is available when archaeologists incorporate ethnoarchaeological case studies. The study of recent examples of megalithic building activities offers the opportunity to study the basic principles and behavioural choices underlying specific megalithic building traditions. By considering appropriate theoretical approaches to these principles and choices, we can develop a model of the social implications of megalith building activities in a specific case study.

Complex phenomena, such as megalithic building activities, are inevitably set in a range of possibilities, spanning from a cooperative and communal appropriation of resources and labour to more exclusionary strategies, implying individualised mechanisms. Further, by considering the possible range of human social organisation, it is possible to compare specific ways of acting and choosing among different societies and communities. This includes theoretical approaches focusing on societal strategies that range from corporate/communal to exclusionary/network courses of action [64–66]. Further, theories on cooperative and collective action provide a basis to describe varying mechanisms that may range between competitive, individualistic, and cooperative actions (e.g. [67–72]). The fluent transition between these different modes of action, the situational importance they may gain within megalithic building activities, and their connection to overarching social structures also offer links to different dimensions of practice theory. Giddens [73, 74] remarks on the dialectic relation between structures and actions. The recursive character of this relation still has importance in archaeological thought (e.g. [75, 76]) and also has potential explanatory power with regard to monumentality (e.g. [77]). Within these wider theoretical frameworks, different types of social organisation and economic strategies can be implicated, thus providing a possibility to describe societies that are very different from each other.

For the analysis and definition of monumentality, aspects of the active shaping and construction of landscape are of special importance. Although monuments may also be natural, non-built features, they are often artificially made. Despite the lack of a clear and singular definition of monumentality, due to the diversity of this phenomena (cf. [63]), one attempt to define monumentality may place a focus on the outstanding role of monumental construction [78]. More specifically, monumental construction involves large-scale efforts around an alteration of the environment, involving large and everlasting features. Further, the social effect and role of monuments is an important dimension for classifying monumentality. The social meaning of a monument may be achieved through multiple smaller investments over a long time (e.g. [79]) or a single, large-scale investment. Connected to the social sphere of megalithic building, the monuments are seen as an important aspect in the creation of social memories (cf. [80, 81]), as well as central places for the interaction of communities. The close entanglement of monumentality and the importance of landscape is visible in the research focus on phenomenological approaches (e.g. [56]), but also on other approaches that highlight the manifold entanglements of monuments and their position within the socially and politically constructed landscape (e.g. [82–84]).

## Results

### Field work and methodology

The results presented here are based on ethnoarchaeological fieldwork undertaken in, among others, the southern part of Nagaland, Northeast India, in February and March 2016 and

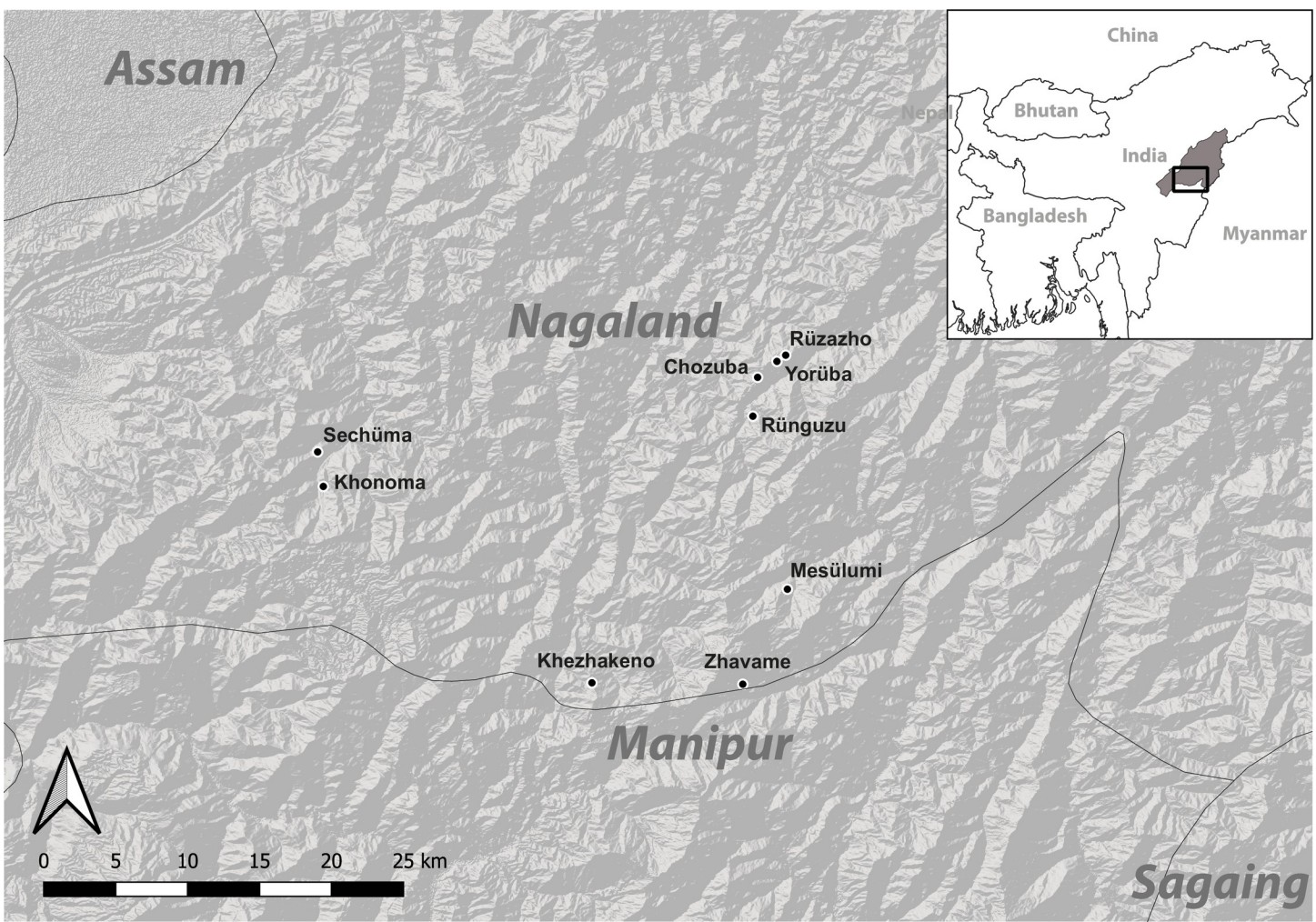

**Fig 3. The villages in Nagaland visited during fieldwork in 2016.** Khonoma and Sechüma are Angami Naga villages; the remainder are Chakhesang Naga villages (Graphic: RGK Frankfurt, M. Wunderlich; shape files made with Natural Earth, DEM based on CGIAR-CSI SRTM data. DEM republished from Jarvis A., H.I. Reuter, A. Nelson, E. Guevara, 2008, Hole-filled seamless SRTM data V4, International Centre for Tropical Agriculture (CIAT), available from http://srtm.csi.cgiar.org, under a CC BY license, with permission from Alliance of Bioversity International and CIAT, original copyright 2004–2021).

March 2018 (Fig 3). Fieldwork was carried out within the framework of the Priority Program "Early Monumentality and Social Differentiation" (German Research Foundation).

Semi-structured interviews were conducted in each village with as many interview partners as possible. Since megalithic building had been abandoned during the 1960s to 1970s, few of the people who had witnessed the feasting and construction activities, or whose parents had been feast-givers, were still alive. The interviews were conducted with the help of a translator, since the inhabitants of the villages spoke different dialects. The interviews were open-ended and often included a group of two to three people providing answers and discussing memories. Due to the very different roles of men and women in the former tradition of feasting and megalithic building activities, the majority of the interview partners were male. However, we did conduct some interviews with women. Women were also sometimes present during the interviews with men and sometimes provided additional viewpoints. The interviews aimed to collect individual perspectives on the specific traditions of the given villages and to complement the data available from ethnographic descriptions (e.g. [13]). All in all, 20 interviews

were conducted, in 11 villages. Prior to the fieldwork, the methodological outline of the project, including semi-structured interview technique, was approved by the Academic Council of Nagaland University, the highest decision-making body in all academic and ethical matters of the university. Furthermore, a detailed outline of the research plan and objectives, as well as the intended methodology, was given in each of the visited villages in presence of the village councils. All research activities, including the planned interviews, were approved by the village councils and consent was given orally. When the interviews took place with either single informants or smaller groups of informants, consent to use the information for analyses and publication was given orally and was witnessed by the research team, and local cooperation partners who served as translators.

The data collection was carried out in a collaborative effort, jointly organised by members of Kiel University and Nagaland University. The documentation focused on villages located in the districts of Phek and Kohima, which are associated with the Angami Naga and the Chakhesang Naga. The research methodology consisted of a comprehensive documentation of the megalithic monuments themselves, as well as the conducting of interviews [3]. The megalithic monuments were documented with regard to their size, type, shape, orientation, and specific location and, where this information was available, their association with individuals or specific social groups within the village. The interviews focused both on different aspects of the monuments themselves (such as reasons for their location) and on the accompanying feasting activities, which are inseparably entwined with megalithic building in Nagaland. Although the possibility of an association between material culture and megalithic monuments was not the subject of the research presented here, there are data available from test excavations from a few villages of the surveyed district. These include data from the villages of Jotsoma [28], Khezhakeno, Khusomi and Movolomi [42], which yielded evidence of iron spear heads, machetes, plain and cord-marked pottery, ground stone tools of sandstone (e.g. at Khusomi), found in association with cist burials, standing stones, sitting platforms, and stone circles. It was also stated during some of the interviews, that plants and/or food was deposited close to the standing stones in the framework of certain feasts.

To answer the questions posed above, we provide general descriptions based on the anthropological literature of the dimensions of kinship, social organisation and feasting activities, referring only to the Angami and Chakhesang Naga, in the southern part of Nagaland. We note that these dimensions may differ from other areas of Nagaland. We complement these descriptions with our own ethnoarchaeological descriptions of one village, which was visited in 2016 by the research team. Since it is a basic characteristic of Naga communities that they represent strongly independent units in terms of economic, political, and social dimensions [85], it is important to describe these dimensions, for example, feasting activities and megalithic building for one specific community.

## Actively shaped kinship

Within archaeological research, the perception and interpretation of kinship is traditionally heavily based on concepts of biological kinship, in which the (biological) lineage is considered the most important unit [86]. Within social anthropology, this rather static notion of kinship first came under critique during the 1970s. Since then, and especially with the formation of 'new kinship studies', concepts and theories that focus on the active formation and negotiation of kinship have been integrated. Although the formal element of genealogical relations was, and still is, seen as one part of kinship structures (cf. [87]), these new approaches focused on the variability and diversity of concepts of kinship beyond consanguinity and affinity in different cultural contexts (e.g. [88–92]). This notion is not undisputed and is set within a wide

discourse that includes very different positions on the actual importance of biological procreation within kinship systems. These positions range from a complete refusal of performative notions of kinship, to rather one-sided positions neglecting any importance of biological relatedness.

A shared characteristic of these studies is their emphasis on the social processes that construct close affiliations and kinship structures. Those may be shaped by specific rules and laws; by shared ritual experiences; by structures of reciprocity, solidarity, and sharing; or by the desire and/or need for a stark differentiation from other groups (e.g. in cases of conflict; cf. [71]). Hence, important referential frames of analysis could be the processes of the social construction, maintenance, and processual (re)assessment of relatedness, which may accumulate and merge within biological relations, symbolic representations, and daily activities (cf. [88]).

In their basic social organisation, especially with reference to kinship systems, southern Naga communities may be described on several distinct levels which are in themselves intertwined through various kinds of relationships and influential dimensions of daily life. The first, or basic, unit is the nuclear family. This family consists of a married couple and their children, who lived together in a house until the children reached a certain age. The second unit is the lineage and clan-system, which was bound together with reference to a common ancestor. Both levels, or units, were defined patrilineally in all Naga groups. Despite this focus on and reference to the paternal side of the family, the maternal side was also considered of great importance within the kinship system. This system included a comprehensive terminology distinguishing between maternal and paternal relatives ([85], 103f.).

Especially the smallest groups within the lineage and clan-systems are defined by a close biological relatedness (as siblings, aunts, uncles, etc.) and usually are of great importance for inheritance rules. Although these groups may be dispersed over several villages, at least a large proportion of its members are usually found within one village ([93], 90f., [94], 124f.). Together, a number of these social groups, whose ancestry can be defined very clearly, may form a lineage. In turn, several lineages may form a clan. Although the smallest groups are the most important for a sense of relatedness and close relationships, it is the clans which serve as the basic functional unit within the villages. Membership in a clan brings with it a complex system of rights and obligations and is thus being created and recreated within specific frames of interaction. An obvious form of these frames of interaction are the different forms of cooperation, in which each and every clan member is expected to participate. These cooperations include a broad spectrum of activities and dimensions, including assisting in agricultural activities (e.g. preparation of the fields for shifting cultivation), construction activities (e.g. house construction), and support in the case of unforeseen events (e.g. disease outbreak or fire). Furthermore, the clan is the foundation of social security in economic and physical matters, providing assistance in case of need. Also, clans usually hold collective land areas, such as cultivated and forested areas ([93], 87–90). Clans also played an important role within the traditional political organisation of the villages. Within the open and non-institutionalised hierarchies of the Angami and Chakhesang Naga, administrative decisions concerning village affairs were made by a selected group of members called *krüta*, comprising *thürimave* (ones who had taken heads), *betchimi* (village elders), *thüvomi* or *thevo* (village priest), feast-givers and clan representatives. In these *krüta*, the only fixed social position was that of village priests (*thüvomi* or *thevo*). In general, priests fulfilled a position of spiritual leadership and played an important role within ceremonies and feasts, especially those which were connected to the agricultural calendar. Despite the importance of the position, it was not always inherited among the Chakhesang Naga, and in some cases new priests were chosen owing to their personal eligibility for this position ([21], 125f.). The remaining men taking place in the process of decision making were successful hunters, warriors, men known for their wisdom, and feast-givers.

Therefore, political and social influence within the villages was determined by the categories of age, personal success and achievement. With this influence, there is an inherent connection to feasting practices and the erection of megalithic monuments, which, in turn, served as a materialisation of the completion of a feast series (see below) ([21], 116–120, [94], 127f.).

Families, lineages and clans have to be characterised as social groups that define their relatedness by a shared ancestry and a reference to a common origin. However, the ties and relatedness especially within the partly huge and spatially dispersed clans are mostly formed and maintained by interaction, obligations and reciprocal relations. Therefore, not only biological kinship, but also a more open concept of social relatedness forms a system of kinship that is strongly based and dependent on an active formation by its members.

Another layer of these actively formed groups are the *khels*. These constitute not only a spatial unit subdividing the villages, but also a social group which was particularly important for daily interaction and communal frameworks. A *khel* as a clearly defined spatial unit could include several different clans, or it could be formed by only one clan. *Khels* had great importance for several dimension of social life because they provided an organisational framework for the allocation of working groups. These groups were organised by age class and had responsibility for different tasks, such as the care for the village gates or specific agricultural activities ([93], 40f.). Probably the most important aspect of *khels* was that they provided a communal framework for the socialisation and education of children. At a certain age, children used to move into a special house, the *morung*, where they received an education and were involved in different collective tasks. Therefore, *morungs* were of fundamental importance for the development and social bonding within the social groups involved ([94], 135f.).

The perception of social relatedness within Angami and Chakhesang Naga communities provides important clues about how a sense of social relatedness may be developed, structured, and maintained. The biological dimension of kinship and the active components of kinship should be seen as equally important for relatedness within these communities. Megalithic building is an important aspect in this regard. As will be shown in the next part of this paper, these construction activities involve important social groups (clans and *khels*) and summarise different mechanisms that are important for the development of social relatedness.

As mentioned in the theoretical section of this paper, social relatedness as a concept always relies on a differentiation between different social groups, and at the same time functions in a unifying way within these groups. Kinship and relatedness are, above all, structuring elements within a societal framework. Among other things, they are based on interdependencies and reciprocal relations. Due to the shared education of children and the manifold cooperation (within working groups), the interdependencies especially among members of the same *khel* were quite high. These connections between the members of a *khel* were not directly linked to a shared origin but, rather, shaped through interaction and reciprocity and thereby created an active network of fundamental importance for the overall social organisation and relatedness.

## Layers of landscape

The structures and units of kinship, together with, economic areas and megalithic monuments, create a unique interplay of different spheres of interaction and agency (Fig 4). The enculturated landscape created by three different and distinct layers separates different social mechanisms and arenas and is also strongly connective in itself.

**The village: A social arena.**   Located within the central area of this landscape is the village itself, constituting the first layer of the landscape. Within the village, naturally, daily interactions and daily life take place. Moreover, it is the space where social relations are defined, negotiated, and maintained. The most important social units are located and interacting within the

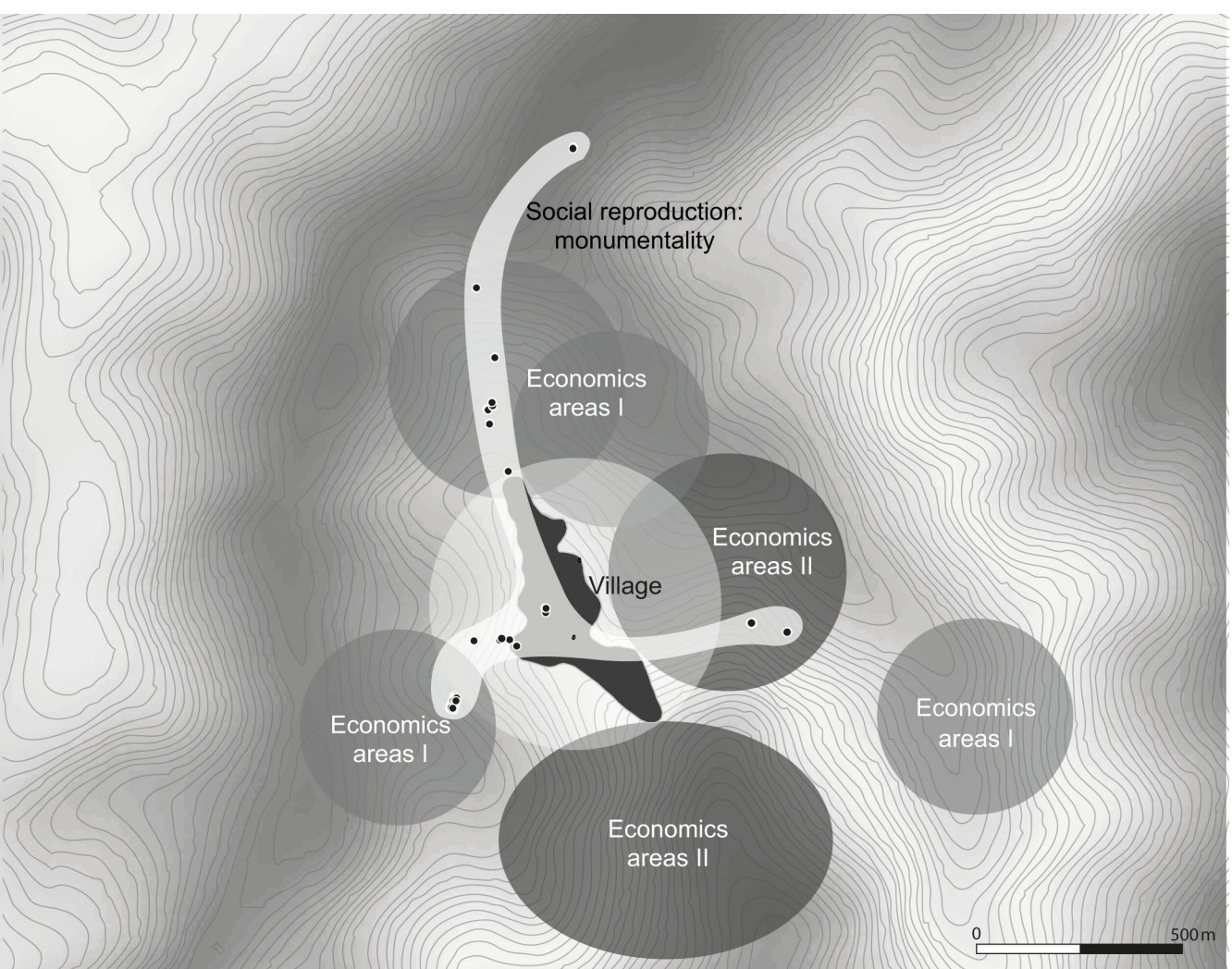

**Fig 4. A model of the different spheres of the socially constructed landscape of Rünguzu (graphic: M. Wunderlich, J. Cordts (graphics department of the Institute of Pre- and Protohistoric Archaeology Kiel); DEM based on CGIAR-CSI SRTM data.** DEM republished from Jarvis A., H.I. Reuter, A. Nelson, E. Guevara, 2008, Hole-filled seamless SRTM data V4, International Centre for Tropical Agriculture (CIAT), available from http://srtm.csi.cgiar.org, under a CC BY license, with permission from Alliance of Bioversity International and CIAT, original copyright 2004–2021).

borders of this village (cf. [85, 93]). The *khels* are themselves spatial units, but they also comprise important social units. Cooperative structures are developed and maintained within the communal houses of the *khels* (i.e. the *morung*). In addition, the function of the *khels* as an interacting and influential social group is materialised through the monumental sitting platforms, which are prominently placed within the central area of the *khels*. Here, social relations, influence, and questions are discussed and actively shaped.

But also the clans are rooted within the village area. As the most important cooperative unit within southern Naga communities, clans have a strong influence over every aspect of social affairs. Further, clans form an important kinship group. Relatedness is actively shaped through participation in feasting activities, communal building activities, and cooperation, for example in, agricultural activities. The importance of clan members to the single individual is shown by their participation in the allocation of resources needed for the feasts of merit. Although the prestige and status connected to the organisation of feasts are mainly attached to the feast-

giver himself, these structures actually provide the required cooperative background in order to enable the collection of the needed resources. Therefore, the monuments should always also be seen in the collective framework, in which the construction of the monument is embedded through the cooperation of, for example, clan members.

Hence, this first layer of the enculturated landscape, the village, is comprised of different factors. First of all, it is the place where kinship and relatedness are created and fostered. It also provides the collective framework in which both the social group and the single households act and are enabled to create and negotiate social status. Lastly, it is the place where communal decision making takes place and where competition and social prestige are materialised through the display on houses.

**The economic areas.** A second layer of the landscape is formed by the economic areas surrounding the village itself. These areas can be divided into two distinct categories, which are influenced by very different characteristics. The first category, the terraced fields, are mainly used for wet-rice cultivation and normally are rather small in size. They belong to single households, hence representing the most important form of individual resource. The second category, the forested areas, are used for different purposes. In many areas of Nagaland, shifting cultivation was used either alone or in combination with terrace cultivation [95]. The areas used for shifting cultivation constitute an important area used for garden plot cultivation and are in many cases created in a collective way. The slash-and-burn activities are mostly carried out cooperatively by the *khels* and clans. Further, the forested areas are used for providing the timber needed for construction, as well as for hunting activities. Lastly, they are also used for cattle herding, which is again connected to individual property.

Subsequently, the economic areas represent two distinct and very important aspects. Especially the terraced fields are the basis for surplus production and economic inequality. The creation and maintenance of economic inequality through the economic areas is a distinct feature that, in turn, influences the structures within social groups, open hierarchies and the importance of single households, as well as social groups within the communities.

**The footpaths.** The third layer of the enculturated landscape is a space with less clear boundaries. The footpaths connecting the village area, as an arena of interaction and kinship structures, and the economic areas, as a basis of economic inequality, are also a distinct feature in themselves. This area is marked by the accompanying standing stones, whose distribution blurs into both the outskirts of the village and the economic areas.

This connecting space has different meanings and functions. First of all, this area serves as a commemorative space, where the accomplishments of single individuals and households are remembered and materialised through megalithic monuments. They are inseparably connected to the feasting activities, although no distinct feasts or rituals take place at the monuments themselves. Besides having a commemorative function, the space between the village and the economic areas is also an arena where social prestige and influence are materialised and actively shaped through the erection of monuments. This refers not only to the builders themselves, but also to the importance of the space used for the dragging of stones from their original location. Here, warriors and other influential and important persons gained a special, accentuated place within the crowd, thus fostering open hierarchies and symbolising social standing. Due to the collective allocation of resources needed for the feasting arrangements, the monuments further relate to the communal structures behind these activities. These are, in turn, influenced by kinship structures and a sense of relatedness, fostering reciprocal structures and interdependencies.

It becomes clear that this model of landscape construction provides a framework in which different modes of action and mechanisms are interconnected with each other and materialised with varying emphases. Megalithic monuments are an interconnecting element of

paramount importance, where superficially opposing concepts, such as reciprocity and competition, as well as individualised and communal frameworks, are merging into each other. All these different layers of landscape and actions are also visible, and influential, in the case of the Chakhesang Naga village of Rünguzu, which will be used as a detailed example of the practices of feasting and megalithic building.

## The Chakhesang Naga village of Rüngüzu

Rünguzu is a village which is part of the Chakhesang Naga and located in the southern part of Nagaland. Surrounded by terraced fields and forested areas, Rünguzu covers an area of 5.6ha (Fig 5).

The village is oriented approximately N-S, with the terraced fields lying mainly to the north and south of the village. Within the village of Rünguzu, two *khels* are present. Phüswünumi *khel* is located in the northern part and Nyingatsomi *khel* in the southern part of the village. They are each inhabited by eight clans, but these clans are not spread among the *khels*. In the Phüswünumi *khel* are present the Vasa/Tunyi, Thorünu, Ngulhünumi, Züvenu, Shijoh, Nakro, Rhakho, and Tetseo clans. In the Nyingatsomi *khel* are present the Lüruo, Nienu, Vero, Medeo, Müswyi, Thisho, Keyho/Veswü, and Kusünumi clans. During the interviews, it was

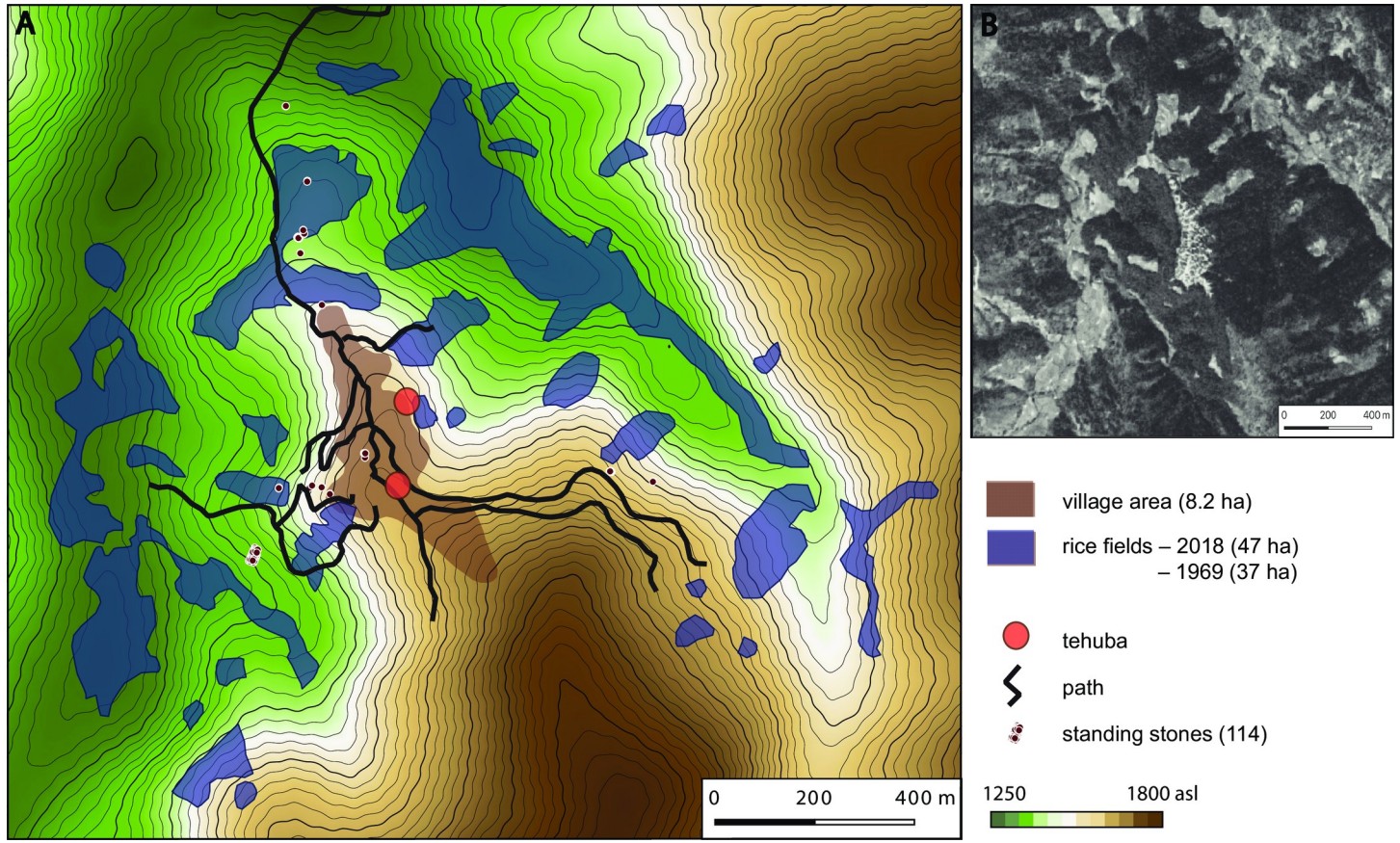

**Fig 5. The Chakhesang Naga village of Rünguzu and its surrounding landscape.** A: data collected during the field season in 2016. The distribution of the standing stones marks the paths leading towards the terraced fields. B: satellite image of the village (Graphic: RGK Frankfurt; B based on CORONA Satellite Photography, (USGS) EROS Center (EDC); DEM based on CGIAR-CSI SRTM data. DEM republished from Jarvis A., H.I. Reuter, A. Nelson, E. Guevara, 2008, Hole-filled seamless SRTM data V4, International Centre for Tropical Agriculture (CIAT), available from http://srtm.csi.cgiar.org, under a CC BY license, with permission from Alliance of Bioversity International and CIAT, original copyright 2004–2021).

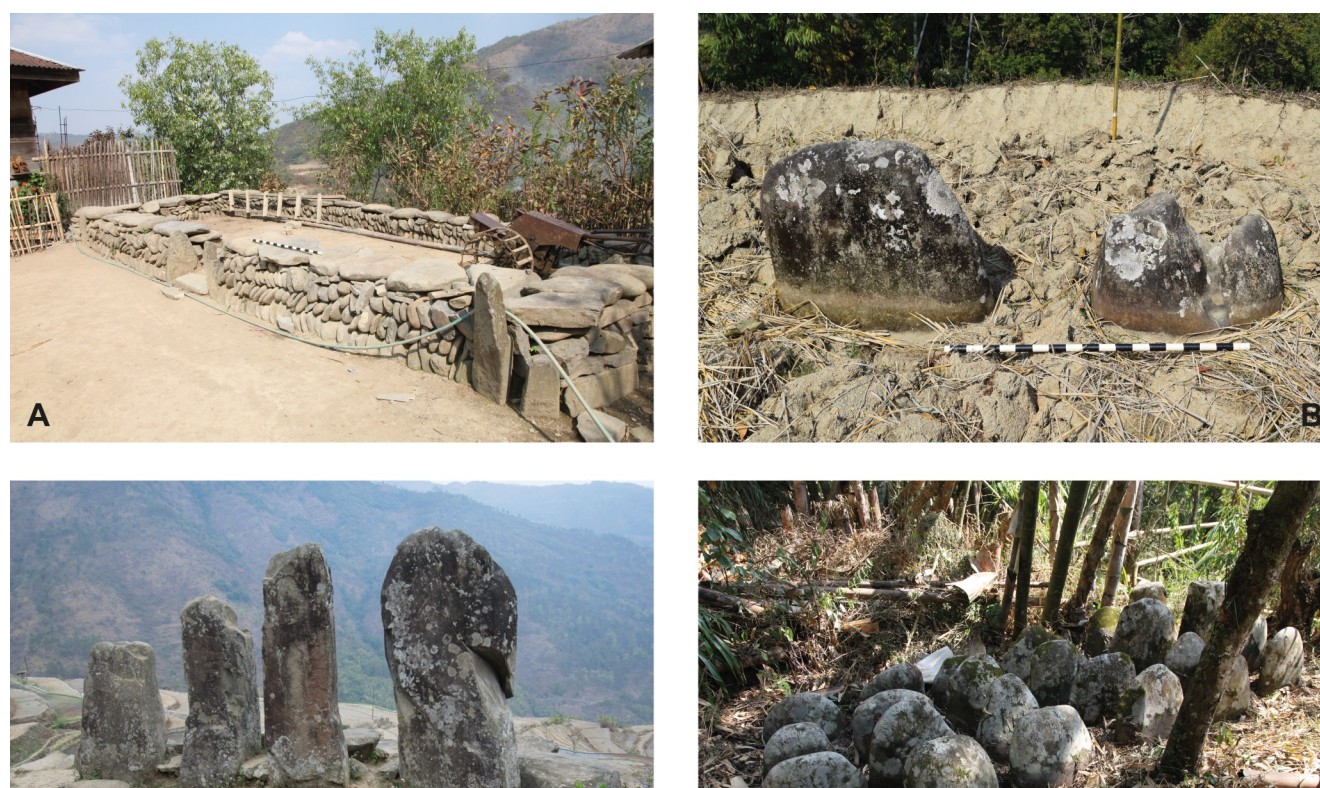

**Fig 6.** The different types of megalithic monuments present in Rünguzu: A: Sitting platform (Photo: K. Rassmann, RGK Frankfurt), B: Standing stones without platform (Photo: M. Wunderlich), C: Stone row with a small stone platform (Photo: S. Jagiolla, graphics department of the Institute of Pre- and Protohistoric Archaeology Kiel), D: Stone field/cluster (Photo: K. Rassmann, RGK Frankfurt).

stated that all the clans from each of the two *khels* play an important role, especially during megalithic construction activities, thus being one of the cooperative mechanisms which fostered a distinct sense of relatedness within these social groups.

**Rünguzu: The stone monuments.** All in all, 20 monuments of different types were documented within the area of Rünguzu (Fig 6). Of these, only a few are located within the village itself (compare Fig 5). Those are the assembly places of the two *khels* of the village and two smaller standing stone monuments. Both sitting platforms are set in a rather central position within the northern, respectively, southern village area. All the other monuments are located near the footpaths leading out of the village area towards the terraced fields. This divergent placement of megalithic monuments is a common trait within the communities of southern Nagaland. The vast majority of the standing stones are found near these paths (compare Fig 5). However, if we take a closer look at the affiliation of the monument builders, it becomes clear that the placement of the standing stones near the fields is not intended to create a separation between the different social groups. On the contrary, the monuments are assorted and can be assigned to the two different *khels* and their different clans (Fig 7).

The monuments themselves show what can be described as a divergent influence of uniformity and individualism with respect to the relatively small overall number of monuments. Among the uniform characteristics are the number of stones and the orientation of the

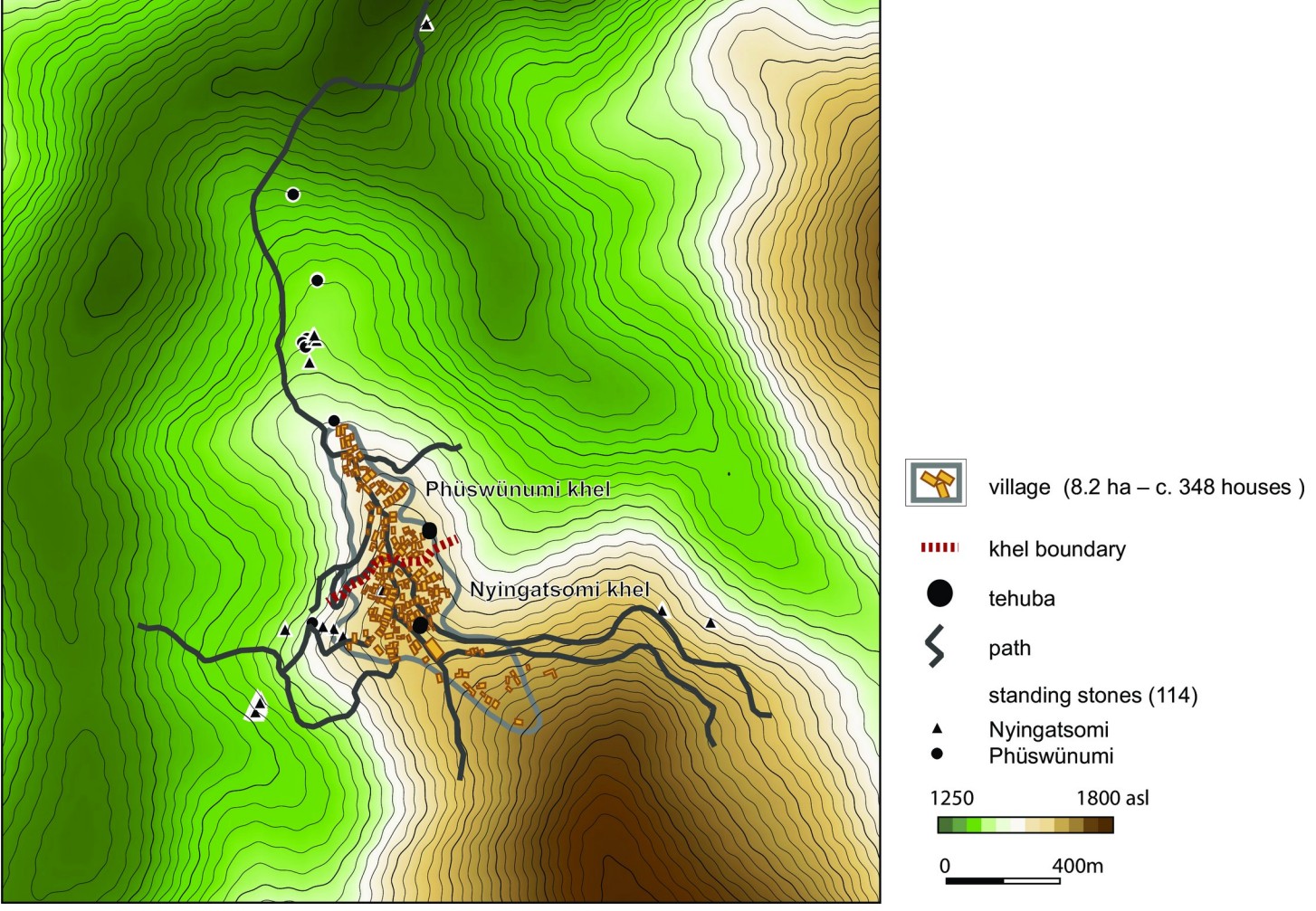

**Fig 7. The association of the megalithic monuments with the two *khels* (Nyingatsomi and Phüswünumi) present in the Chakhesang village of Rünguzu.** The two *tehubas* within the village area belong to one *khel* each (Graphic: RGK Frankfurt, M. Wunderlich; DEM based on CGIAR-CSI SRTM data. DEM republished from Jarvis A., H.I. Reuter, A. Nelson, E. Guevara, 2008, Hole-filled seamless SRTM data V4, International Centre for Tropical Agriculture (CIAT), available from http://srtm.csi.cgiar.org, under a CC BY license, with permission from Alliance of Bioversity International and CIAT, original copyright 2004–2021).

monuments and the stones themselves. Both factors are governed by certain rules, which are connected to specific social rules and allow only a slight variation with reference to the orientation of the monuments, which ranges between N–S, NE–SW, and NNE–SSW. The rather uniform character of the monuments is also visible in the fact that only three different main monument types are present in Rünguzu. The first main type is the already mentioned sitting platforms, which occur two times and are assigned to the two different *khels* of Rünguzu (Fig 8). The second main type are rows of stones, which can be subdivided into rows with and without an accompanying stone platform (compare Fig 6). These make up the biggest part of the dataset (n = 14 of 20) and can be further differentiated according to the number of stones included (see below). The last main type is the monuments, which are characterised by several rows, or a smaller fields of standing stones. This type occurs four times in Rünguzu.

One of the individualised aspects of megalithic building is the number of stones per monument which varies greatly within the Rünguzu dataset. Megalithic monuments in Rünguzu consist of either 2, 4, 8, 20 or 32 stones. These numbers are consistent with the rules, and each

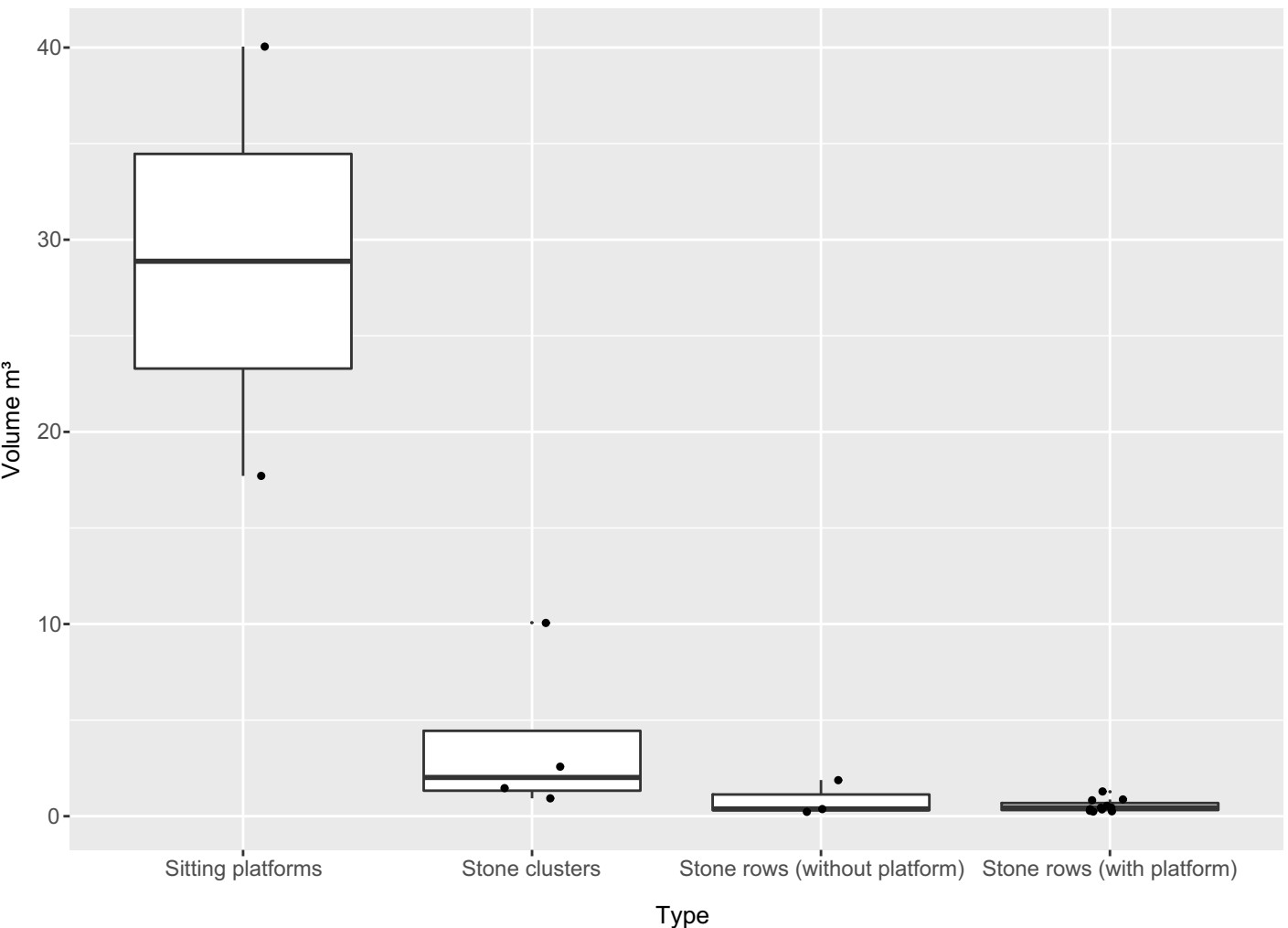

**Fig 8. The four different types of megalithic monuments and their size distribution (m³) in the Chakhesang Naga village of Rünguzu (M. Wunderlich).**

and every time a monument erection is held, both a male and a female stone should be erected, thus resulting in an even total number of stones. Therefore, the smallest monuments in Rünguzu consist of two stones. Most of the monuments (n = 11) consist of two stones, although four monuments are built up of four (n = 3) or eight stones (n = 1). The remaining three megalithic monuments are much larger. Two monuments of 20 stones and the previously mentioned monument of 32 stones complete the dataset in Rünguzu. This largest monument, with its high number of stones, is an exception. In Rünguzu, but also in other villages, the number of stones per monument was always described as being of greater importance than the actual size (m³) of the stones.

As already mentioned, megalithic building among the Angami and Chakhesang Naga involved not only the feast-giver's family, but also additional persons related to the family. Therefore, the monuments are also, to a certain degree, connected to these overlying collective structures. To this respect, a clear imbalance is visible in the dataset. A rough division can be made between the monuments belonging to the two different *khels* within Rünguzu (Figs 9 and 10). Most of the monuments (n = 14) can be assigned to the first *khel*, Phüswünumi. Besides the sitting platform, these monuments comprise all of the different monument types

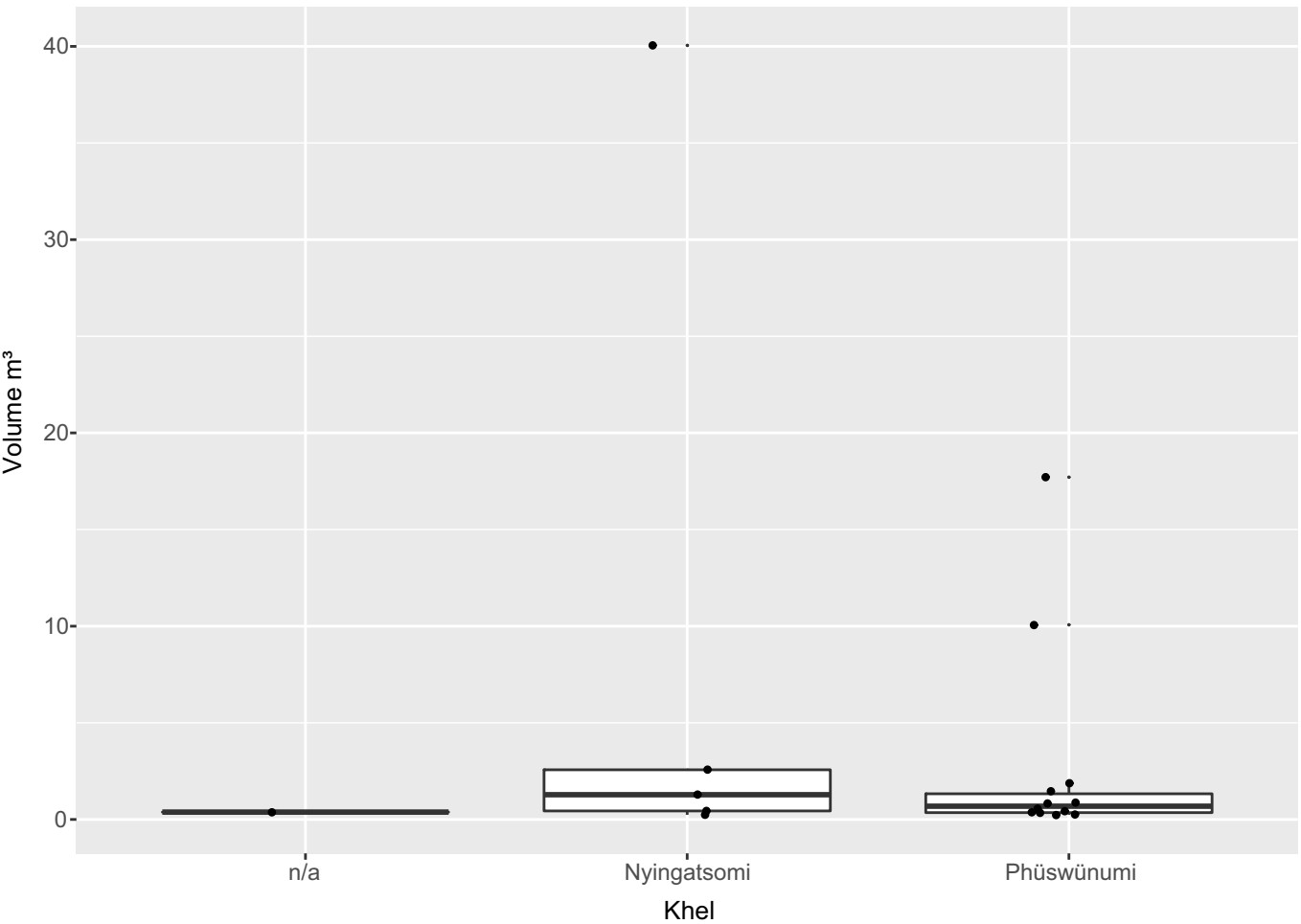

**Fig 9. The size of the different monuments per *khel* in the Chakhesang Naga village of Rünguzu (graphic: M. Wunderlich).**

present in Rünguzu with a rather strong emphasis on stone rows without platforms. Only five monuments can be assigned to the second *khel*, Nyingatsomi: three stone rows, a single field of stones, and a sitting platform.

The second basic level of social organisation, the clans, must also be mentioned in this regard (Figs 11 and 12). As mentioned above, altogether, eight clans can be assigned to the Phüswünumi *khel* and eight to the Nyingatsomi *khel*. Only three members of two different clans (Müswyinumi, Lüruo) of the Nyingatsomi *khel* built megalithic monuments. This includes three stone rows without a platform (2×2 stones, 1×4 stones) and one field of stones (20 stones). A much higher number was built by the different clans belonging to the Phüswünumi *khel*. Here, most of the megalithic monuments were erected by members of the Ngulhünumi clan (n = 6). The monuments include one stone field (8 stones), two stone row with a platform (4 and 2 stones), as well as three without a platform (all 2 stones). Three further monuments were erected by members of the Thorünumi clan, all of them being stone rows without a platform (2×2 stones, 1×3 stones). Members of the Vasa clan built two monuments, one of them being a stone row without platform as well (2 stones) and one being one of the biggest stone fields, with 20 stones. Lastly, one monument each can be assigned to members of the Züvenumi and Nakro clans. The stone field belonging to the Züvenumi clan is the biggest

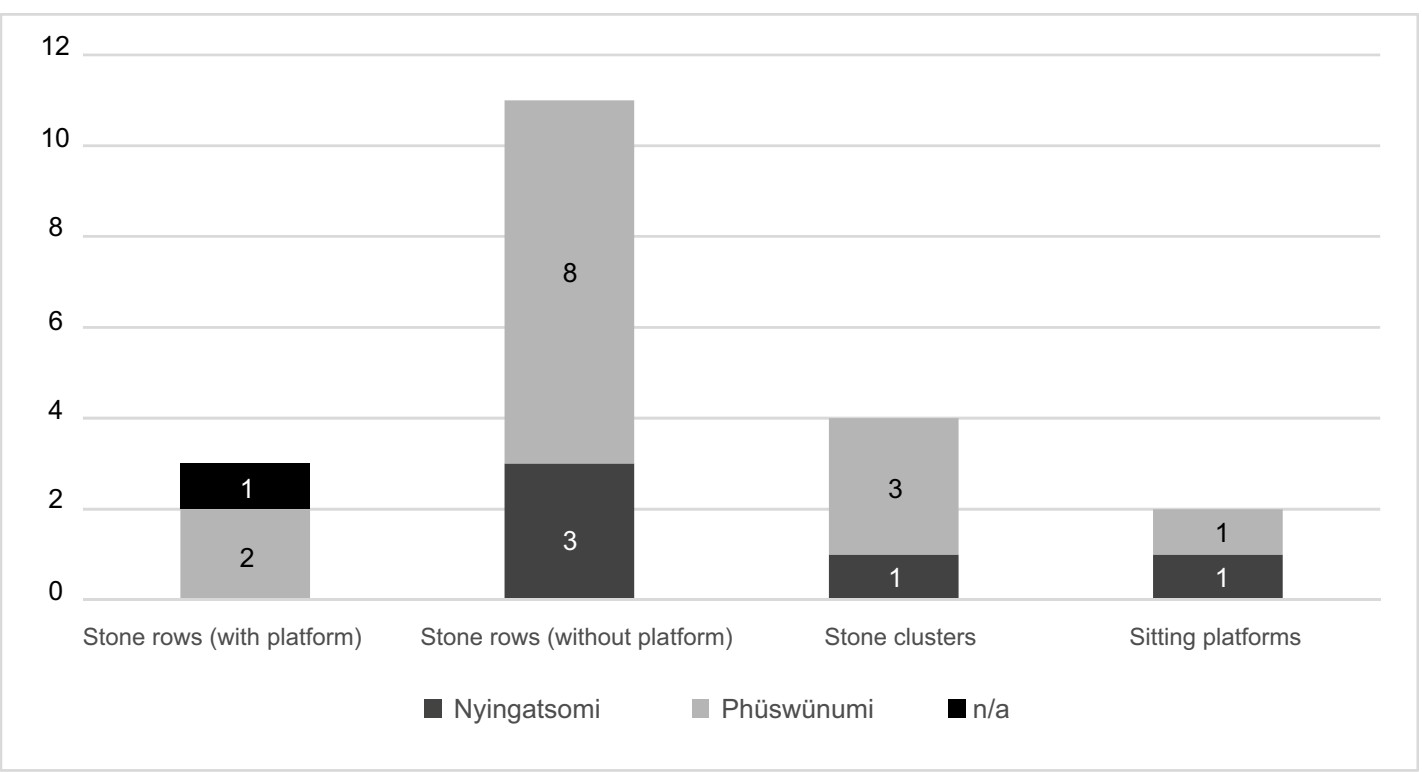

**Fig 10. Diagram of the different monument types per *khel* in the Chakhesang Naga village of Rünguzu (graphic: M. Wunderlich).**

monument in Rünguzu and contains a total of 32 erected stones. The monument of the Nakro clan is a stone row without platform and again includes two stones.

**Rünguzu: Feasts of merit.** As it is a common and conjunctive element of megalith erection in southern Naga societies, the entire process of megalithic building is connected to a specific series of feasts. In general, the so-called feasts of merit are a series of different feasts, which may be started after a man has founded his own household and started to cultivate his own rice fields. The first feasts are usually rather small in scope, including mostly only a small amount of resources and involving a smaller group of participants. The subsequent feasts grow consecutively bigger, involving greater amounts of resources and greater numbers of participants to be invited. The resources required are rice, pigs, cattle, and sometimes *mithun* (*Bos frontalis*).

During the interviews, it was always stated that the feasts of merit constitute an expensive and hard-to-achieve accomplishment. Exact statistics on the percentage of village members who achieved the different stages of the feasts remain to be determined, although it was stated in one village that roughly 30 percent managed to complete at least some of the feast stages. In general, the feasts of merit constituted an important way to achieve influence and prestige within Angami and Chakhesang communities. With the completion of the feasting series, individuals gained the right to participate in the process of decision making within the village. In contrast to the generally shared and conjunctive concept of feasting and megalithic building, the feasts of merit follow a unique, individualised, and detailed structure in each and every village (cf. [13, 96]).

In Rünguzu, the feasts of merit comprise a series of three feasts, which may be executed in a simple or a complex manner. In the simpler version, *Süna*, in the first feast, the feast-giver may

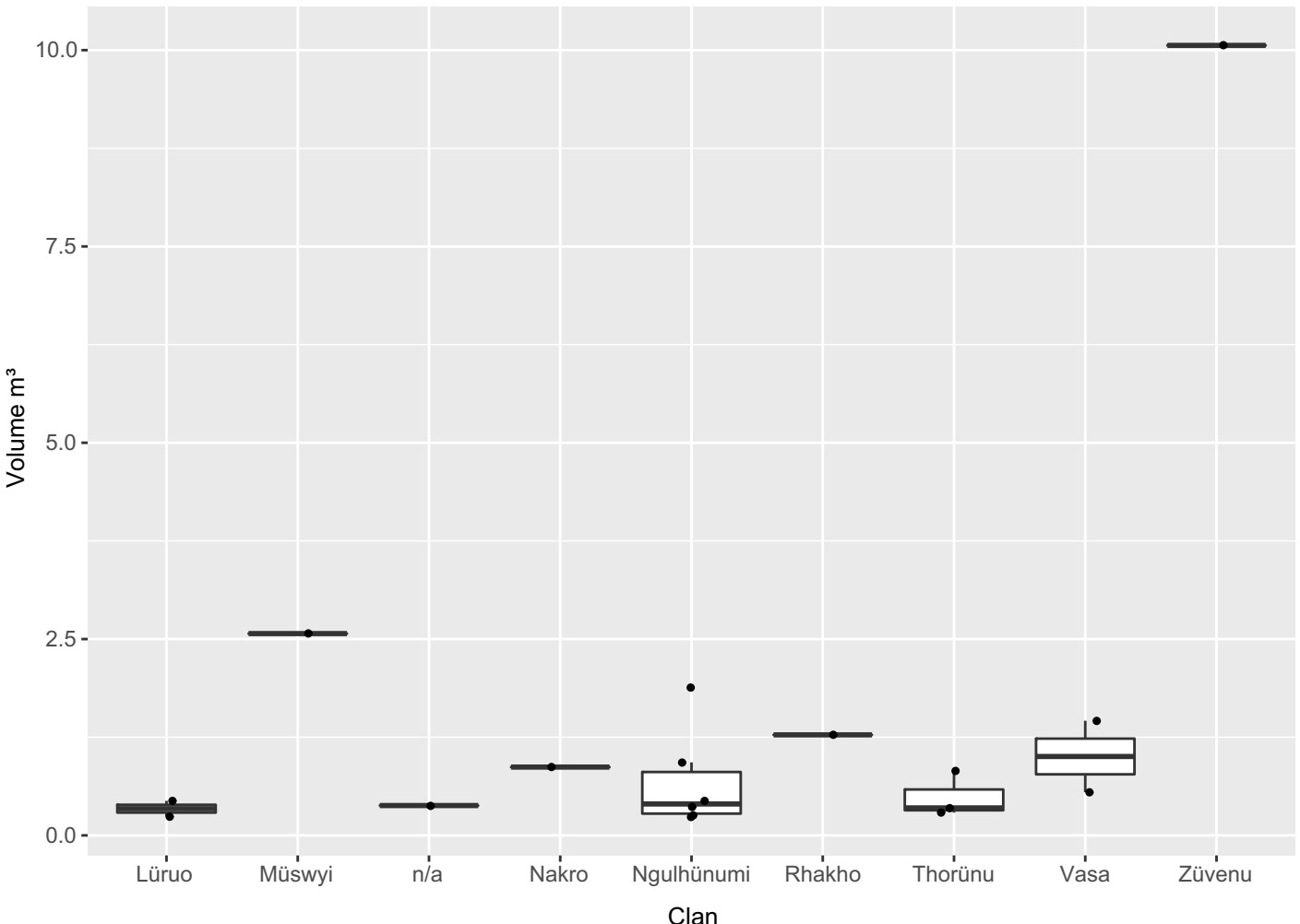

**Fig 11. The size of the different monuments per type belonging to the various clans in the Chakhesang Naga village of Rünguzu (graphic: M. Wunderlich).**

attach a house decoration (house horn) to their house and in the second feast, the feast-giver may drag and erect two stones. In the complicated version, *Tünyena*, the feast-giver was required to give at least two feasts before the first two stones could be erected. In both versions, the completion of the entire ritual required 11 independent feasts, each of which could be held within a timespan of one year. The number of stones dragged and erected was 2 stones for the second and third feasts, 4 stones for the fourth and fifth feasts, 8 stones for the sixth and seventh feasts, 12 stones for the eighth and ninth feasts, and lastly 16 stones for the tenth and eleventh feasts. After the completion of this entire series, if the feast-giver/individual still wished to host another feast, he was given a symbolic rebirth by being dressed up like a small boy and carried in a basket to begin the entire series again (i.e. he would be reborn).

The exact number of animals to be slaughtered for the different feasts very much depended on the number of people invited. For this, there was no definite rule described in Rünguzu, but invitations expanding beyond the two *khels* in Rünguzu naturally greatly increased the number of animals needed for slaughter. As a minimum needed for each feast, our informants gave a number of three to four pigs and two cattle, and they stated that this number had to increase from feast to feast.

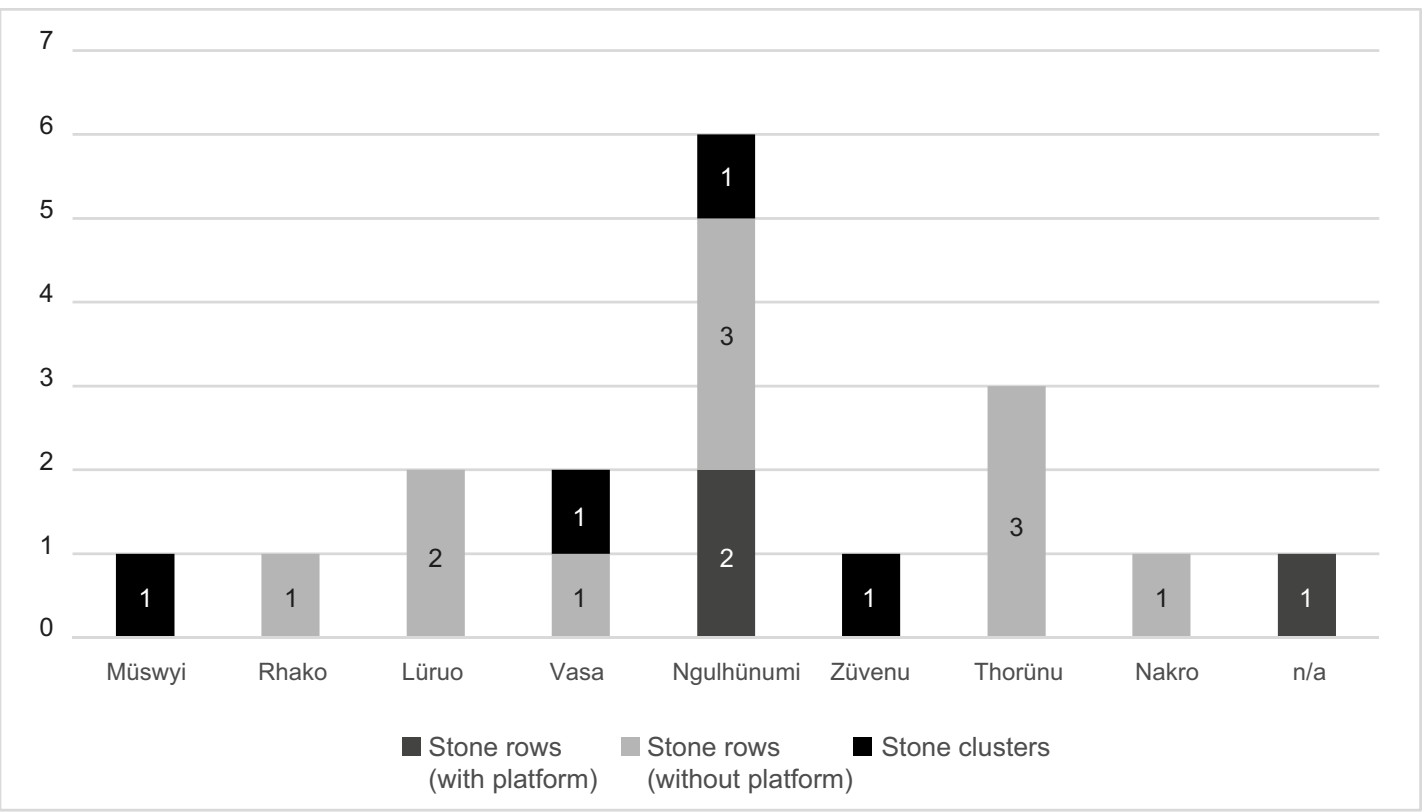

**Fig 12. Diagram of the different monument types per clan in the Chakhesang Naga village of Rünguzu (graphic: M. Wunderlich).**

The erection of the megaliths themselves also followed a defined process. The stones had to be chosen by the feast-giver and associated persons from a communally held area near a river located 2–3km away from the village. The stones remained in this place until the next day. During the following night, the persons involved in the choice of the stones would wait for dreams telling them the names of the stones. During both the *Süna* and the *Tünyena* feast, the stones would be transported from the river to the place where they were to be erected, with the participation of the male members of either the respective *khel* or the entire village. During the dragging process, which made use of a wooden sledge, a prominent position was given to head-hunters and warriors. After the stones had been dragged to their final destination, they were erected that same day, with the help of ropes and wooden poles. Within the enculturated landscape, the standing stones served as memorial places for the feast-giver and his status. Within the village, this was expressed through the right to wear a specific shawl known as *thüpi khwü*, as well as the attachment of a *ceka* house horn.

Quite interestingly, during the interviews, stones were described as agents. They were seen as living persons that themselves acted throughout the dragging journey, by supporting or rejecting the direction in which the men were taking them.

## Representations through materiality

Feasting activities and the investment of economic resources are materialised through different materials and find their reflection both in wood and in stone. The erected stones themselves are to be seen as the last step of the feasting series. The houses also show representations of feasting activities. House horns are a symbol for the completion of earlier stages of the feasts of

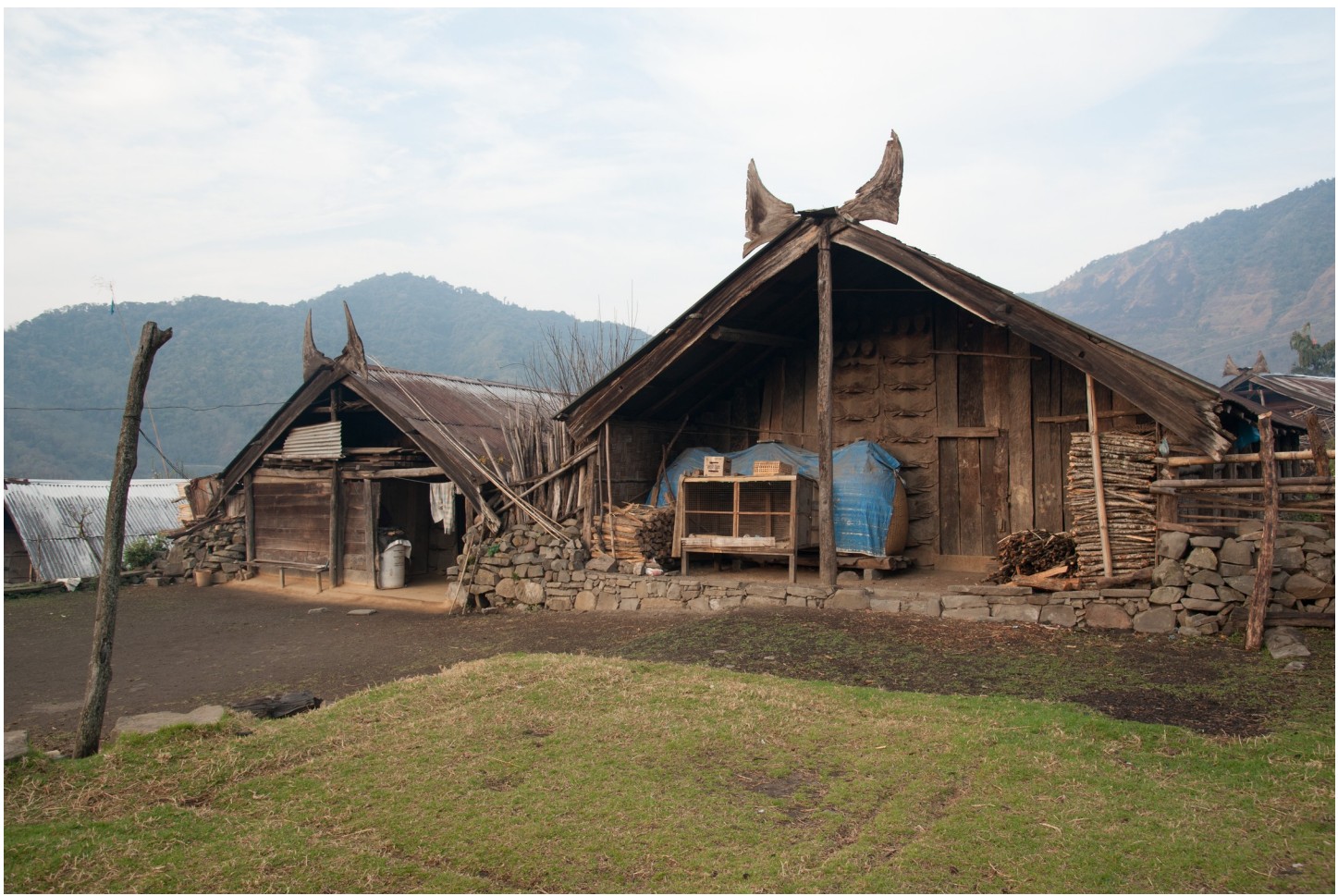

**Fig 13. Two older houses in the Chakhesang Naga village of Zhavame with house horns, marking the completion of a certain stage of feasts of merit (photo: S. Jagiolla, graphics department of the Institute of Pre- and Protohistoric Archaeology Kiel).**

merit (Fig 13). Depictions of *mithun* heads are to be found on the house fronts, thus creating a very direct and obvious connection between a single household and the resources invested into feasting activities (Fig 14). Lastly, the front posts of the houses are decorated and carved in specific designs as well.

The representation of activities that are of communal importance also stretches to the long since-abandoned practice of head-hunting. In Zhavame, a Chakhesang village, several head stones could be documented (Figs 15 and 16), representing the heads taken by members of the village community. The carved head stones together form two separate installations, which are in close spatial proximity to each other (distance between the two clusters of head stones: ca. 50cm). Within each installation, the head stones are erected immediately adjacent to each other and therefore must be considered as one monument, although the individual head stones are unique with regard to their appearance. These monuments are located outside of the village area, along one of the footpaths leading to the terraced fields. They are located directly below the ending of a stone avenue of standing stones, which ends at a break-off edge into the valley. The special position at this spot can be explained by the fact that it is precisely here that the view opens into the valley and onto the terraces. The descending or ascending person is thus shown the practice and success of head-hunting in a form materialised in stone. As they are

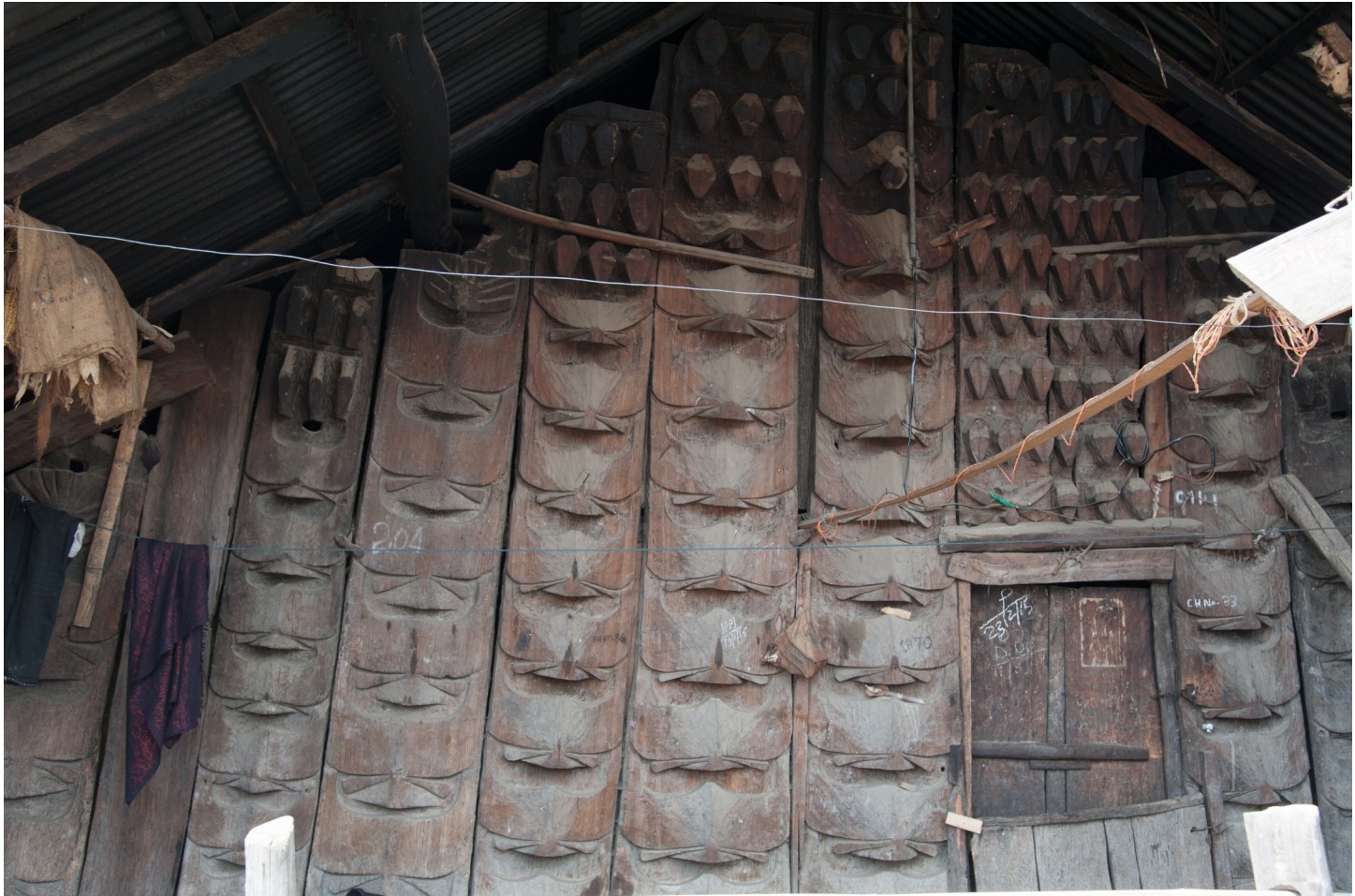

**Fig 14. A house front in the Chakhesang Naga village of Zhavame with the traditional carvings, including *mithun* and human heads, still preserved (photo: S. Jagiolla, graphics department of the Institute of Pre- and Protohistoric Archaeology Kiel).**

not directly surrounded by the standing stones that accompany major parts of this footpath until the termination of the stone avenue in higher altitude distance 100m; they therefore form a distinctive feature just next to the path.

In conclusion, these head stones were located outside of the village area, near the beginning of the long and impressive stone avenues leading towards the village. This position within the socially constructed landscape is of great significance, since it provides a signalling element to anyone entering the village. A stylistically very similar equivalent is found both on house fronts, and on the village gates (Fig 17).

The reflection of feasting activities, as well as head-hunting shows the interlinked patterns and levels of social interaction and individual efforts and prestige. Although the practice of head-hunting should be seen as a collective framework, it is materialised and depicted not only on the communal village gates, but also on individual house fronts. As described earlier, megalithic building is set in an interesting entanglement of individual action as well as collective references.

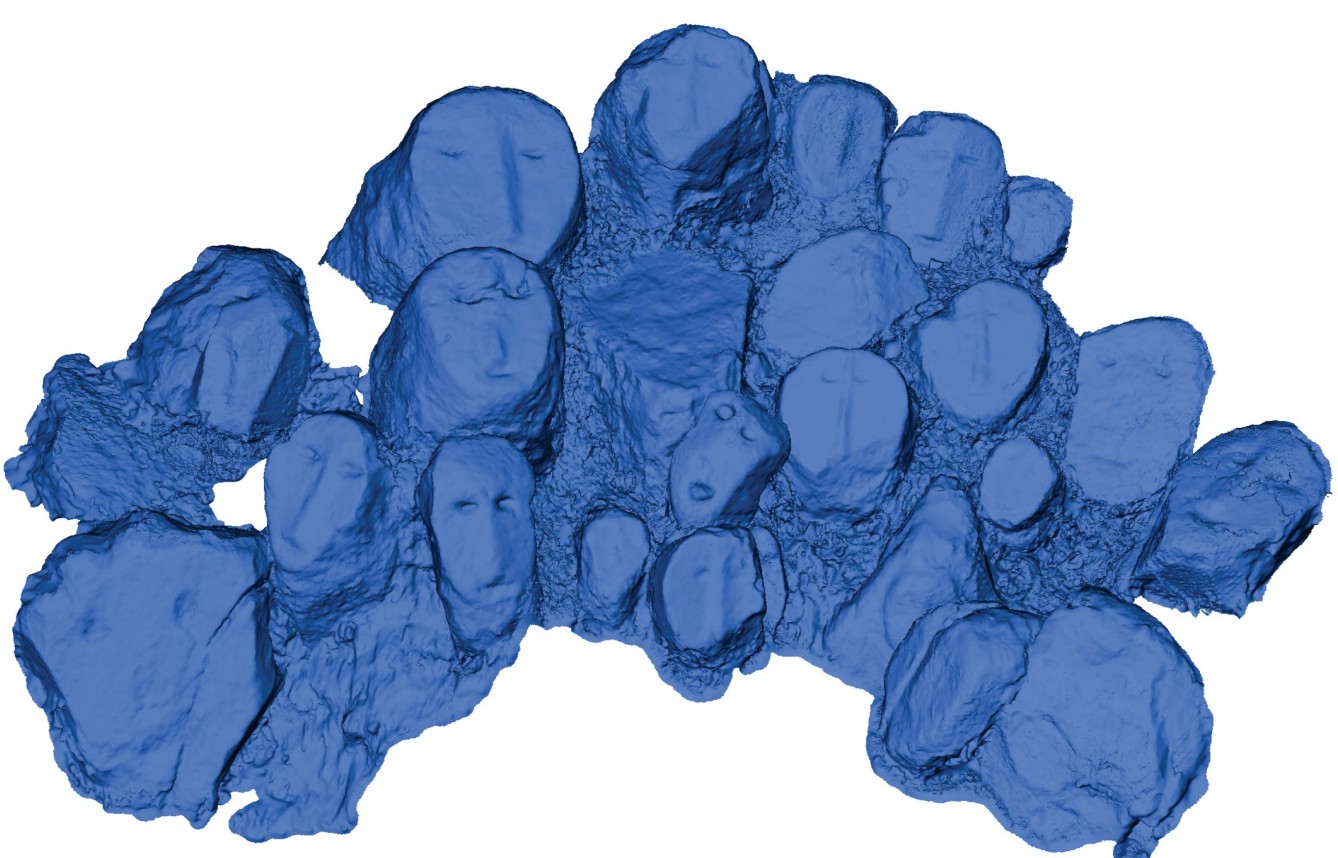

**Fig 15. Model of the carved head stones at the footpaths in the Chakhesang Naga village of Zhavame (model: S. Jagiolla, graphics department of the Institute of Pre- and Protohistoric Archaeology Kiel).**

## Discussion

A description of feasting activities, as well as monumental building strategies, among southern Naga communities should necessarily include a comparative view. Although the feasts of merit, as well as the erection of megalithic monuments, are rooted in a shared idea or strategy of societal organisation, there are already striking differences visible on the level of different villages. The villages visited during the field work are set within an area of 1100 km$^2$ and show a strong entanglement with each other. This is, for example, visible in the distribution of clans, which may reside in multiple villages and therefore create strong networks. Within all these places, the underlying principles of feasting practices and megalithic building are the same: they follow the same goals and share the same underlying mechanism. Still, each village shows its own execution, arrangement, and emphasis.

For the megalithic monuments, these differences can be detected in the varying occurrence and differentiation of monument types (Table 1). The most striking difference can be seen with regard to the occurrence and importance of sitting platforms. Within the two Angami villages, the sitting platforms constitute 24 and 31% of the total number of megalithic monuments. Within these villages, sitting platforms may be erected both by collectives (*khels* and clans) and by individual families. This is in sharp contrast to the Chakhesang villages, where only four out of eight villages had any of these platforms at all. Only one village, Chozuba, has a high percentage (36%) of these monuments, whereas in the other villages the percentage of sitting platforms ranges between 1 and 14%. The size of the sitting platforms reflects these

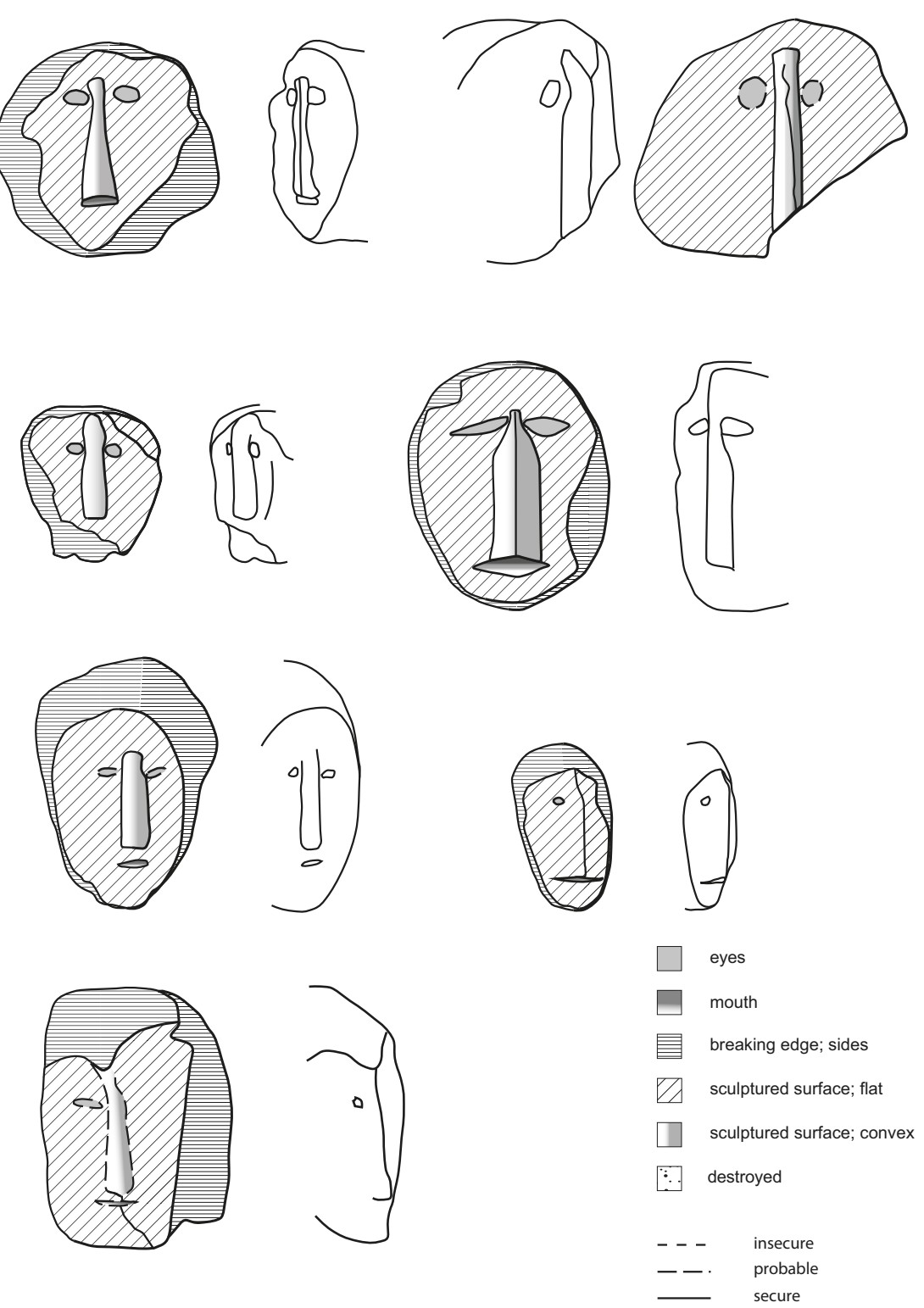

**Fig 16. Technical drawing of selected head stones in the Chakhesang Naga village of Zhavame (drawing: S. Beyer, graphics department of the Institute of Pre- and Protohistoric Archaeology Kiel).**

patterns; the volume (m$^3$) of Chakhesang sitting platforms is usually in the lowest spectrum compared with Angami sitting platforms (Fig 18).

With regard to the second and third main types of monuments, standing stones and stone rows with and without attached platform, the differences within the datasets are present throughout the Angami and Chakhesang villages (compare Table 1).

Single standing stones may not be present at all (this is the case in the two Chakhesang villages) or may range between 3 and 81% of the total number of monuments. The size of the single standing stones is, for most villages, quite equally distributed and ranges between 0 and 5m$^3$ (Fig 19). Still, some villages show remarkably big monuments of this type, with a volume of up to 58m$^3$. The size of this monument type is mainly influenced by the attachment of stone platforms, which are especially common in the village of Khezhakeno.

A similar, but slightly different, picture is visible concerning the stone rows. This type of monument is present in all the villages included here, although the percentage is again highly variable, ranging between 16 and 74%. As it is the case with the single standing stones, the size of the stones rows is mainly influenced by the addition of stone platforms. These platforms are the main influence on the size distribution per village (Fig 20). Here, the vast majority of monuments fare up to 10m$^3$, while the biggest monument of this type reaches a size of over 50m$^3$.

This overview of the comparative data sets shows, how diverse the phenomenon of megalithic building is even within a small, local environment. The differentiation between the villages is comprehensible with regard to the types that villagers chose to build, the size of the monuments, and the overall composition of monument types. Still, the underlying mechanisms and choices that influence megalithic building activities were strikingly similar in all the villages.

As described above, specific practices of feasting as well as megalithic building constitute an important element of the structuration of social relations and open, fluid hierarchies. Within the southern Angami communities, where inherited, fixed social positions were uncommon, these practices were one of the very few ways to gain individual social prestige and a right to participate in and decide on village matters. At the same time, both feasting activities and megalithic building were set within a cooperative framework in all the villages. The cooperative framework mainly concerned the allocation of resources, which was repeatedly described as being organised and resumed by fellow members of the clan and the *khel* of the individual feast-giver. Thus, on the one hand megalithic building is the materialisation of convergent dimensions, but on the other hand, it reflects the individual memorisation of the feast-giver's accomplishments, as well as competition for social prestige and influence. On the one hand, of course, these factors were influenced by a substantial degree of economic inequality, which was closely connected to the ownership of land and livestock. On the other hand, though, cooperative and communal structures were highly relevant both for the feasts of merit and for megalithic building. The resources required often exceeded the possibilities of individual households, thus raising the need for collective action within social groups. Such factors as reciprocal structures, (individual and collective) reputation, and reward systems (cf. [69]) are an important basis for any kind of cooperative and collective behaviour. A high degree of cooperation was achieved through a high degree of interdependency within the main social groups (the clans and *khels*), as well as established debt and solidarity systems within southern Naga communities. As a last factor, specific mechanisms of social signalling seem to have been important for the collective framework of megalithic building. As P. Roscoe ([71], 99) stated,

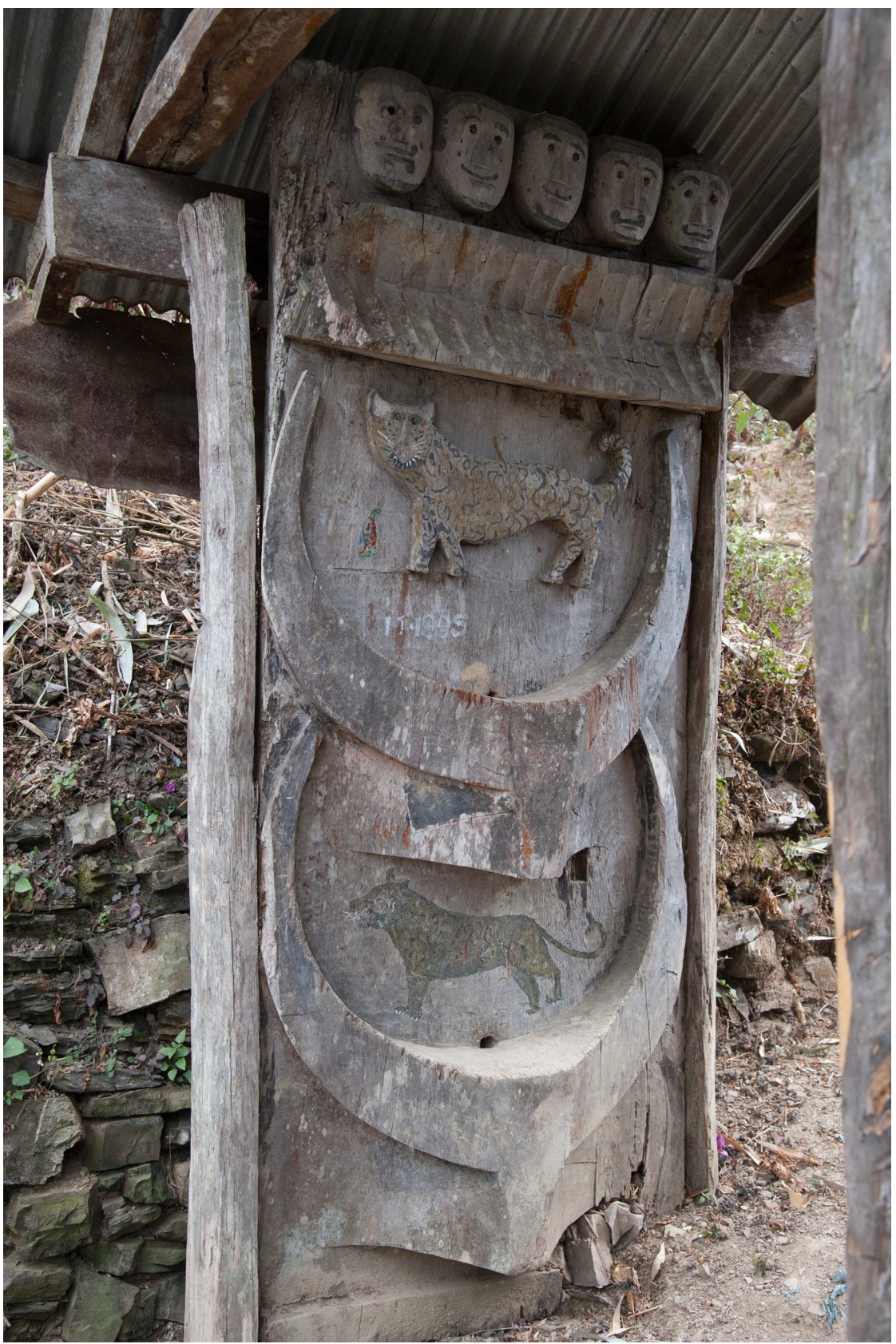

**Fig 17. Decoration on a village gate at the paths leading from the village towards the terraced fields in the Chakhesang Naga village of Rünguzu, depicting human heads and *mithun* (photo: S. Jagiolla, graphics department of the Institute of Pre- and Protohistoric Archaeology Kiel).**

"*Material, performative, and architectural displays were, as signaling theorists have termed it, indexically related to the qualities they signaled. As direct objectifications of a group's size, and the commitment, coordination, and capabilities of its members, it was simply impossible for individuals or groups to mount a superior display if they did not, in fact, possess these qualities [. . .]*". Megalithic monuments as prominent architectural features within the socially structured landscape of southern Naga communities can be connected to signalling behaviour since they were seen by any outsider approaching the village.

Although this basic, but highly important and influential, mechanism was shared among the different villages, the exact execution and arrangement of feasting and megalithic building activities varied greatly. This points towards the importance of individual translations or embodiments of a shared idea within communities, which are located within a local environment.

The practice of megalithic building within southern Naga communities is comprehensively characterised by various dimensions. First, despite its connectivity with the feasting practices and the gain of social prestige of one single household, it is interwoven with influential collective units, namely the clans and the *khels*. Through the collective allocation of resource and the embeddedness within communal frameworks provided by these units, megalithic building is set within a collective framework. Here, group interests and the signalling of a group's strength in terms of economic inequalities play a significant role. These factors translate well into the behavioural mechanism described as influential for cooperative behaviour, including such facets as reciprocity and reward systems (e.g. [69]). The groups involved in this cooperative and communal framework show the importance of kinship and relatedness within the practice of megalithic building. In addition, both of the main units of social organisation, the clans and the *khels*, clearly show the entanglement of relatedness and social space. Clans, with their clear reference to a common ancestor, are characterised by a high degree of interdependency and

**Table 1. The number and percentage of the different monument types in villages visited in 2016.**

| Village | Sitting platforms | Single standing stones (without platform) | Single standing stones (with platform) | Stone row (without platform) | Stone row (with platform) | Stone cluster | Various | n/a |
|---|---|---|---|---|---|---|---|---|
| **Khonoma (n = 120)** | 24/24% | 20/17% | 11/9% | 24/13% | 27/23% | | 1/1% | 13/13% |
| **Sechüma (n = 23)** | 7/31% | 2/9% | 2/9% | 1/4% | 9/39% | | | 2/8% |
| **Chozuba (n = 39)** | 14/36% | 1/3% | | 16/40% | 1/3% | 2/5% | 1/3% | 4/10% |
| **Khezhakeno (n = 119)** | 2/1% | 58/49% | 37/32% | 3/3% | 16/13% | 1/1% | | 2/1% |
| **Khusomi (n = 6)** | | 1/17% | | 3/50% | | 2/33% | | |
| **Mesülumi (n = 54)** | | 7/13% | 5/9% | 16/29% | 19/35% | 5/9% | 1/2% | 1/3% |
| **Rünguzu (n = 21)** | 2/10% | | | 11/52% | 3/14% | 4/19% | | 1/5% |
| **Rüzazho (n = 41)** | | | | 26/63% | 2/5% | 10/24% | | 3/8% |
| **Yorüba (n = 31)** | 4/14% | | 1/3% | 23/74% | | | 1/3% | 2/6% |
| **Zhavame (n = 69)** | | 10/14% | 38/55% | 4/6% | 8/12% | | | 9/13% |

Khonoma and Sechüma are Angami Naga villages; the remainder are Chakhesang Naga villages.

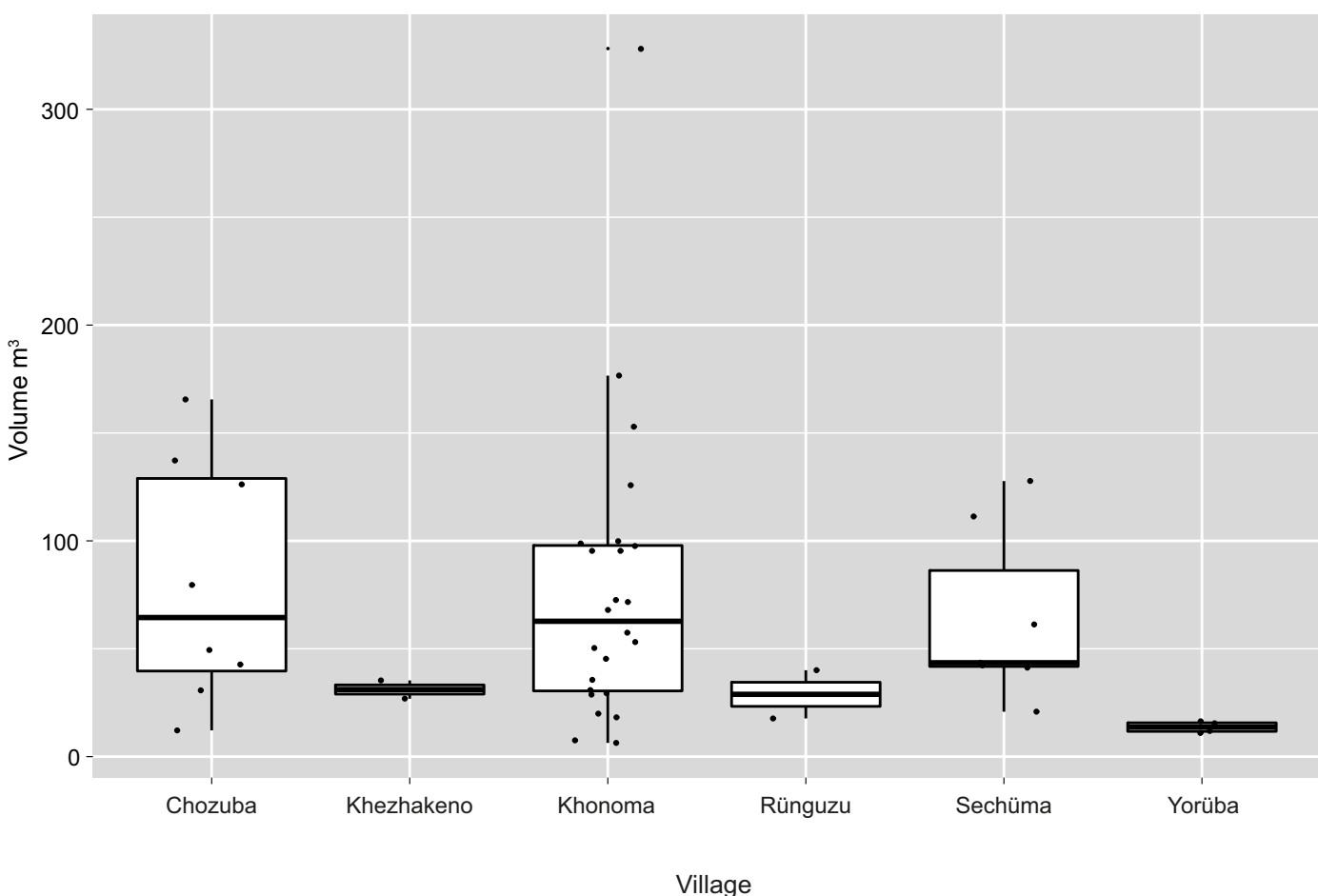

**Fig 18. The size distribution (m³) of sitting platforms in the different villages visited in 2016.** Khonoma and Sechüma are Angami Naga villages; the remainder are Chakhesang Naga villages (Graphic: M. Wunderlich).

further connect several villages with each other, thus creating networks of kinship structures beyond the village unit. The *khels* are important for the structuration of the social space within the village. They shape and bound groups of people, also beyond the limits of one clan, and create a framework of communal relations. Here, a sense of closeness, and also of relatedness, is achieved and maintained through mutual aid, interdependencies and collective actions. A materialisation of these relations is, as mentioned above, to be found in the sitting platforms, which are erected by each *khel* of a village and serve as an important gathering place. The importance of factors of social space, and its potential to form residential patterns, has also been stressed by Hamberger [97] with an emphasis on the importance of gender.

A consideration of the deeply intertwined relation between socially constructed kinship groups, cooperative efforts, economic inequality and megalithic monuments also allows the socially structured landscape within southern Naga communities to be understood. The different elements of the landscape are interwoven within a framework, whereby each of the dimensions mentioned above is materialised. The different influential modes of action, such as communal strategies and competitive behaviour, are physically interconnected yet set within different areas. Within this socially constructed landscape, megalithic monuments play a key role because they are a materialisation of all these dimensions and therefore a connective element.

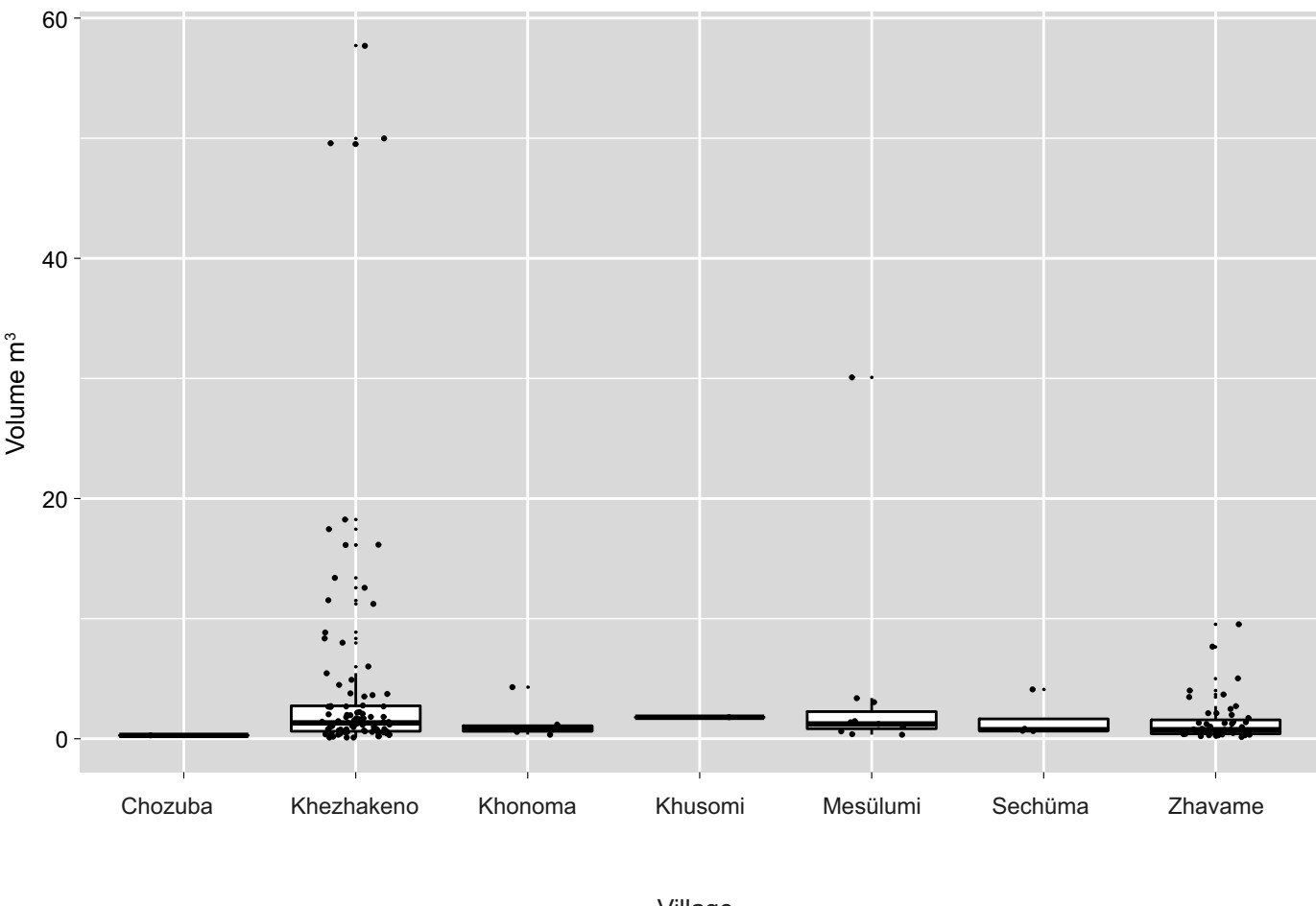

**Fig 19. The size distribution (m³) of single standing stones in the different villages visited in 2016.** Khonoma and Sechüma are Angami Naga villages; the remainder are Chakhesang Naga villages (Graphic: M. Wunderlich).

## Conclusion: Current perspectives from ethnoarchaeological approaches

One of the key results of our inquiries is the observation that an interesting dichotomy exists between shared ideas of a conceptualised use and function of feasting activities and megalithic building, on the one hand, and individual perceptions and alterations of these, on the other hand. Within the investigated Naga communities, these shared concepts are clearly visible as a stable framework, in which the same courses of action and implications of both traditions took place. Still, each and every village showed a remarkably independent translation of monumental building and feasting activities, being differentiated by nuances or clear patterns. The lack of clear-set institutional rules among the investigated Naga communities may be responsible for the great variety in the materialisation and execution of feasting and megalithic building activities.

Megalithic building itself proved to be a meaningful trajectory for the gain of social prestige and status, as well as for the integration of individuals into overarching social networks. The importance of these networks in connection with specific modes of the distribution of resources underlines a very egalitarian approach towards monumentality. The cooperative and

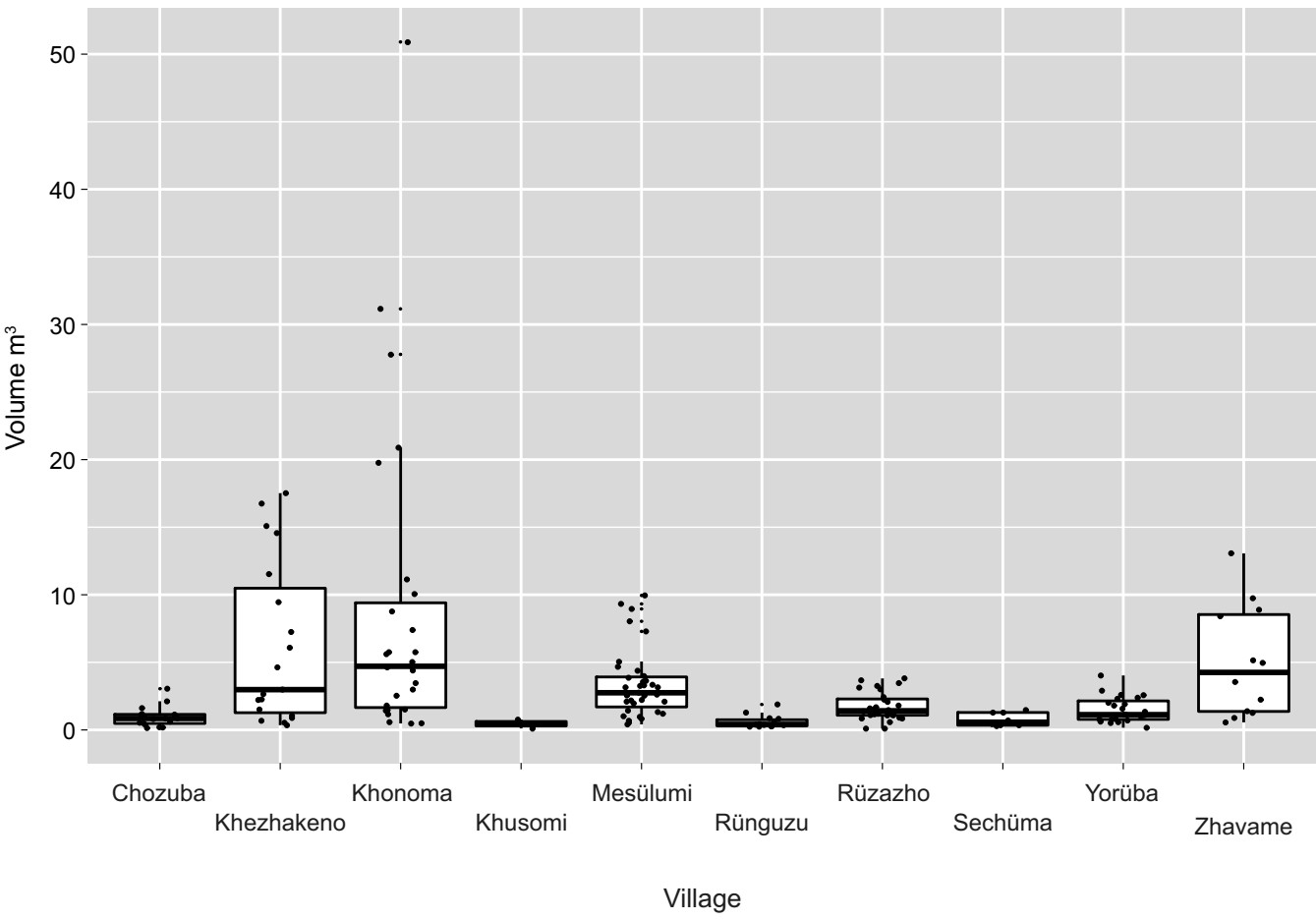

**Fig 20. The size distribution (m³) of stone rows with and without attached platforms in the different villages visited in 2016.** Khonoma and Sechüma are Angami Naga villages; the remainder are Chakhesang Naga villages (Graphic: M. Wunderlich).

also competitive approach towards the erection of megalithic monuments thus allows variation, but no significant differentiation (for example in the size of the erected stones). However, this does not mean that social complexity is not expressed by megalithic building activities. The embeddedness of individuals in specific social networks and their access to the necessary means such as (material) resources, are vital to their ability to start, and proceed with, the feasts of merit. Not only personal agency, but also structural preconditions, which are linked to the social complexity in Angami and Chakhesang communities, influence megalithic building activities. Despite these factors, there were no restriction on feasting activities and megalithic building that were linked to social hierarchies.

In this regard, a future task will be to conduct ethnoarchaeological surveys in areas of highly stratified Naga societies in order to compare the expression of monumentality among societies that are organised in very different ways.

From the viewpoint of European prehistoric archaeology, the results may be considered comparable to observations in different regions exhibiting megalithic building traditions within the Neolithic and Chalcolithic phases. Although a general pattern of megalithic construction is viable, for example, in the Neolithic period in what is now southern Scandinavia and northern Germany, the variation in basic concepts indicates more egalitarian practices and not necessarily the existence of an institution of stratified social practices (cf. [3, 62]).

Applying a bottom-up approach within ethnoarchaeological fieldwork enables a broader focus on the individual agency of communities. The case of Nagaland shows how different the translation of an overarching and possibly uniting tradition such as megalithic building can be. With respect to archaeological data, improved dating techniques and the availability of extensive data sets enable a shift in perspectives towards the individuality of overarching concepts within local communities. Thus, the perspectives provided by an anthropologically informed archaeology can help to open up discourses and analyses that are rooted in the knowledge of material manifestations and, most importantly, the social dynamics and mechanisms behind the archaeological record.

## Acknowledgments

We would like to express our gratitude to the communities in Nagaland, in particular, the Village Councils, and the elderly who shared the traditional knowledge of their respective villages with the team. Thanks are also extended to Dr. Marco Mitri and Prof. H.J. Syiemlieh for their assistance during visits to villages in the Khasi and Jaintia Hills, Meghalaya. We further thank the three anonymous reviewers for their comments and suggestions, which helped us to improve the manuscript.

## Author Contributions

**Conceptualization:** Maria Wunderlich, Ditamulü Vasa.

**Data curation:** Knut Rassman.

**Funding acquisition:** Johannes Müller.

**Investigation:** Maria Wunderlich, Tiatoshi Jamir, Johannes Müller, Knut Rassman, Ditamulü Vasa.

**Methodology:** Maria Wunderlich, Johannes Müller.

**Project administration:** Tiatoshi Jamir, Johannes Müller, Ditamulü Vasa.

**Visualization:** Maria Wunderlich, Knut Rassman.

**Writing – original draft:** Maria Wunderlich.

**Writing – review & editing:** Tiatoshi Jamir, Johannes Müller, Knut Rassman, Ditamulü Vasa.

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
