## [Decision Letter · Decision Letter 0]

12 Nov 2020

PONE-D-20-31416

Societies in Balance: Monumentality and feasting activities among Southern Naga-communities, Northeast India

PLOS ONE

Dear Dr. Wunderlich,

Thank you for submitting your manuscript to PLOS ONE. After careful consideration, we feel that it has merit but does not fully meet PLOS ONE’s publication criteria as it currently stands. Therefore, we invite you to submit a revised version of the manuscript that addresses the points raised during the review process.

All comments must be fully addressed before re-submission.

We look forward to receiving your revised manuscript.

Kind regards,

Peter F. Biehl, PhD

Academic Editor

PLOS ONE

Additional Editor Comments:

Your manuscript has now been seen by three referees, whose comments are appended below. You will see from these comments that while the referees find your work of great interest, they have raised some concerns that must be addressed before re-submission.

Journal Requirements:

3. We note that Figures 1, 2, 3, 4, 5, and 17 in your submission contain map and satellite images which may be copyrighted. All PLOS content is published under the Creative Commons Attribution License (CC BY 4.0), which means that the manuscript, images, and Supporting Information files will be freely available online, and any third party is permitted to access, download, copy, distribute, and use these materials in any way, even commercially, with proper attribution. For these reasons, we cannot publish previously copyrighted maps or satellite images created using proprietary data, such as Google software (Google Maps, Street View, and Earth). For more information, see our copyright guidelines: http://journals.plos.org/plosone/s/licenses-and-copyright.

(1) You may seek permission from the original copyright holder of Figures 1, 2, 3, 4, 5, and 17 to publish the content specifically under the CC BY 4.0 license. 

Reviewers' comments:

Reviewer's Responses to Questions

**Comments to the Author**

1. Is the manuscript technically sound, and do the data support the conclusions?

Reviewer #1: Yes

Reviewer #2: Yes

Reviewer #3: Yes

2. Has the statistical analysis been performed appropriately and rigorously? 

Reviewer #1: N/A

Reviewer #2: Yes

Reviewer #3: N/A

3. Have the authors made all data underlying the findings in their manuscript fully available?

Reviewer #1: Yes

Reviewer #2: Yes

Reviewer #3: Yes

4. Is the manuscript presented in an intelligible fashion and written in standard English?

Reviewer #1: No

Reviewer #2: Yes

Reviewer #3: Yes

5. Review Comments to the Author

Reviewer #1: This is a fascinating and overall excellent paper, that is hampered in places by poor and unclear writing. I would strongly recommend the authors have a native english speaker or copy editor carefully review it before resubmission.

That said, the content is excellent and fascinating, and I think this manuscript will be of wide interest to researchers interested in megaliths, social complexity, feasting, and archaeological theory more broadly. The concluding argument seemed very resonant with "practice theory" approaches examining the relations between structures and behavior and that might be referenced.

A few more specific comments/suggestions:

1, The sections "Stones connected to specific actions and/or social roles" is very cursory and I had to read it multiple times to understand what “this last group” was meant to refer to. you may want to expand the discussion of what this group is with some examples. (I was confused if you were referring to the previous paragraph or introducing a new kind of monument)

2. p. 11 line 238: Despite the lack of a clear and singular definition? (abundance seems the opposite of what your trying to say here)

3. word use: use of the word “aspects” throughout: “dimensions” may be a more appropriate term in many places.

Mechanism – is also used frequently; should be plural in most of the contexts it is used.

4. Finally, I would recommend moving the section "The Village: A social arena" before your detailed discussion of megalith distribution and feasts of merit. The discussion of megaliths and feast is very powerful and effective and is the main focus of the article. I think setting up the village layout as the background to discussing megalith building and feasts would better build the argument and leave a stronger impression of the main issues with readers.

Overall this is an excellent article and with minor revisions and a lot of copy editing will be an excellent contribution

Reviewer #2: This article is a highly informative documentation of the megalithic constructions in one particular village in NE India in conjunction with interviews with informants regarding individual and collective feasting activities in which monumental construction was embedded in the context of a relatively non-hierarchical social structure. The parameters of the study are presented in detail as are the social practices associated with megalithic construction and physical details of the constructions themselves. This will be a valuable ethnoarchaeological case study of megalithic building practices in a relatively non-hierarchical society.

It should be published with just a few minor revisions:

Given the prominence of sitting platforms with in terms of their frequency as well as size and social significance, I would very much like to see at least a representative example illustrated.

I would also have liked to have seen photographs of the other types of stone rows with and without platforms, stone clusters, etc.

This is especially true for the "head stones". The models and drawings of the "head stones" are fascinating, even if they do not figure into the argument of the article to a great degree. But the illustrations left me confused. Were they part of a single installation as implied by the digital model, or were they separately located? Where exactly were they located?

Regarding the location of the sitting platforms, head stones, and other types of megalithic constructions--it would have been useful to show the location of these in figures 4, 5, and 17. As it stands, it is difficult to discern which types are located where.

Similarly, would it be possible to demarcate on the maps the spatial locations of the two khels of the village? What is a "tehuba"? It is not defined in the text as far as I can see. Is this a sitting platform?

These are minor points that would make this really good article stronger and even more informative.

Reviewer #3: The manuscript titled “Societies in Balance: Monumentality and feasting activities among Southern Naga communities, Northeast India” is indeed an interesting read. Northeast Indian megaliths are well known for its diversity and as a living tradition (until recently) among many communities. People’s memory associated with these monumental structures need to be recorded as the older persons in a community who have witnessed or were involved in the construction are decreasing in number with the passage of time. Twenty interviews conducted in Nagaland among some of these people (both male and female) have provided some interesting facts about the linkages between feasting and megaliths and the associated complexity of social and political contexts in the Naga society, particularly the kinship and its association with megalithic building. Three important questions have been raised and the authors seek answers using ethno-archaeological methodologies. The following points in the paper need clarification and attention of the authors:

Page 5: “The remains of extensive megalith building activities can be found all over the different areas of Northeast-India … built by the Khasis, the Nagas, the Garos, as well as the Kuki-Chin-Mizo-groups … states of Meghalaya, Nagaland and Manipur ...” Karbis and Tiwas living in Assam and Meghalaya are also known for erection of megaliths of varied shape and size. The recent volume by Q. Marak on Megalithic Traditions of North East India may be referred for further details.

Page 5: “A common characteristic … megalith building is a recent phenomenon which was almost exclusively abandoned within the last decades due to the rise of Christianity”. Several groups/communities are still involved and continuing with the tradition in certain pockets of Northeast India, i.e. Karbis (see K. Choudhury and D. Bezbaruah) and in Manipur (see P. Binodini Devi).

Page 6: “Although many aspects, such as the age and origin of these traditions ...” The sentence “While in some societies exhibiting recent traditions megalith building traditions these are linked to mortuary practices…” needs modification. Recent archaeological investigations conducted in Nagaland may briefly be included for an idea about the problems of determining the age and providing an estimate of the time range of the megaliths in question.

Page 23: Mithun (bos frontalis) should be Bos frontalis (in ittalics)

Page 38: “The cooperative … allows variation, but no significant differentiation (… size of the erected stones).” This is particularly important to note that besides social complexity involved in the tradition, there is not much of difference in the size of the erected stone structure.

Based on the observations made in the paper, can any generalisation be made regarding the form and type of the structures applicable for megaliths found in the neighbouring areas of Northeast India?

Is there any other archaeological material like pottery, stone and metal artifacts observed in the surveyed villages for understanding the nature of habitation and possible links with the megaliths?

6. PLOS authors have the option to publish the peer review history of their article (what does this mean?). If published, this will include your full peer review and any attached files.

Reviewer #1: No

Reviewer #2: No

Reviewer #3: No

---

## [Author Response · Author response to Decision Letter 0]

27 Jan 2021

Answers to comments by reviewer 1

Concerns about language

The paper underwent a proofreading and copy-editing process. 

The concluding argument seemed very resonant with "practice theory" approaches examining the relations between structures and behavior and that might be referenced.

This perspective is now included in the subchapter “The archaeology of megalithic monuments and bottom-up perspectives” (page 12 and 13; line 267ff.). We agree with the reviewer, that practice theory does indeed provide points of connection with the theoretical background chosen for this paper. Most importantly, this field does offer links between two factors which proved to be of high importance in the presented case study: the individual capabilities to organize feasting activities and megalith building activities and their embeddedness into social and political networks, as well as the overall influence of the given social structures of the community involved. These two broader factors are of course interconnected and stand in a recursive and reciprocal relation to each other. Although these points are now highlighted in the article, an in-depth review of the variable and extremely wide field which can be subsumed under the term “practice theory” would go beyond the limits of this paper. In our opinion, it would indeed be a topic for an independent paper to bring together the different strands of interpretation under the larger umbrella of ‘bottom-up’ and ‘practice theory’ approaches. This could indeed constitute a valuable contribution to the overall context of interpretational frameworks of megalithic building activties; but within the scope of this paper such an approach would be too expanding. 

1. The sections "Stones connected to specific actions and/or social roles" is very cursory and I had to read it multiple times to understand what “this last group” was meant to refer to. you may want to expand the discussion of what this group is with some examples. (I was confused if you were referring to the previous paragraph or introducing a new kind of monument)

This section (page 10, line 213ff.) was completely reformulated in order to make the content and examples clearer. 

2. p. 11 line 238: Despite the lack of a clear and singular definition? (abundance seems the opposite of what your trying to say here)

Yes, this true and many thanks for this hint. The term is corrected from ‘abundance’ to ‘lack’

3. word use: use of the word “aspects” throughout: “dimensions” may be a more appropriate term in many places. Mechanism – is also used frequently; should be plural in most of the contexts it is used.

The use of the words “aspects” and “mechanism”, and potential alternatives, was critically reviewed in the course of the English proofreading and copy-editing process. 

4. Finally, I would recommend moving the section "The Village: A social arena" before your detailed discussion of megalith distribution and feasts of merit. The discussion of megaliths and feast is very powerful and effective and is the main focus of the article. I think setting up the village layout as the background to discussing megalith building and feasts would better build the argument and leave a stronger impression of the main issues with readers.

The section was moved into the Results section (page 21 onwards) in order to build up a framework for the following detailed descriptions of the village Rünguzu. For this purpose, minimal alterations of the text were undertaken. Due to this change of the text structure, the reference list had to be altered and updated and is now to be found with a new numeration. Furthermore, the numeration and order of figures had to be changed in accordance with the altered structure of the manuscript.

Answers to comments by reviewer 2

Given the prominence of sitting platforms with in terms of their frequency as well as size and social significance, I would very much like to see at least a representative example illustrated.

I would also have liked to have seen photographs of the other types of stone rows with and without platforms, stone clusters, etc.

One more figure (fig 4) was introduced which contains photos of the different monument types. These include an example of a sitting platform, a row of stones with platform, a row of stones without platform, as well as a stone field. With this, all types represented in Rünguzu are now covered with figures in the article. The former Figure 7 was integrated into this new figure; therefore, the numeration of the figures had to be changed accordingly. 

This is especially true for the "head stones". The models and drawings of the "head stones" are fascinating, even if they do not figure into the argument of the article to a great degree. But the illustrations left me confused. Were they part of a single installation as implied by the digital model, or were they separately located? Where exactly were they located?

More detailed information regarding the installation itself, as well as the location was added on page 33 (line 723ff.). 

Regarding the location of the sitting platforms, head stones, and other types of megalithic constructions--it would have been useful to show the location of these in figures 4, 5, and 17. As it stands, it is difficult to discern which types are located where.

Due to the clustered location of the monuments, the depiction of the monument types on the map is not possible without the integration of several detailed maps. We decided against this measure, because the monument types are completely mixed and don’t show priorities in their distribution. The only types where this is the case are the standing stones vs. sitting platforms (tehuba) and these are already marked on the different maps. Another reason why we decided against such a detailed depiction is that the main differences with regard to the standing stones lies in the number of stones per monument. Again, an integration of the number of stones per monument in the maps would not be possible in a suitable way. 

The headstones themselves are from a different village and can therefore not be included in the maps. 

Similarly, would it be possible to demarcate on the maps the spatial locations of the two khels of the village? 

A marked line between the areas of the two khels in Rünguzu was added in Figure 7.

What is a "tehuba"? It is not defined in the text as far as I can see. Is this a sitting platform?

The term tehuba indeed refers to sitting platforms; it is the Angami-term for this kind of monument. In order to avoid confusion, the term is now being introduced on page 9 (line 180); with the description of the general type ‘sitting platforms’

Answers to comments by reviewer 3

Page 5: “The remains of extensive megalith building activities can be found all over the different areas of Northeast-India … built by the Khasis, the Nagas, the Garos, as well as the Kuki-Chin-Mizo-groups … states of Meghalaya, Nagaland and Manipur ...” Karbis and Tiwas living in Assam and Meghalaya are also known for erection of megaliths of varied shape and size. The recent volume by Q. Marak on Megalithic Traditions of North East India may be referred for further details.

The subchapter “Megalithic monuments in Northeast India” (page 5ff.) was expanded. Additional information and references were added with regard to the geographical distribution of these monuments. The introduction of the following subchapters now also contains a first overview of the variations and variability which is connected to the megalithic monuments of NE-India. 

Page 5: “A common characteristic … megalith building is a recent phenomenon which was almost exclusively abandoned within the last decades due to the rise of Christianity”. Several groups/communities are still involved and continuing with the tradition in certain pockets of Northeast India, i.e. Karbis (see K. Choudhury and D. Bezbaruah) and in Manipur (see P. Binodini Devi).

This is of course true; more details and further references are provided on page 6 (line 128ff.). The respective communities, in which megalith building is still practiced despite the mentioned societal changes are briefly mentioned now. 

Page 6: “Although many aspects, such as the age and origin of these traditions ...” The sentence “While in some societies exhibiting recent traditions megalith building traditions these are linked to mortuary practices…” needs modification. Recent archaeological investigations conducted in Nagaland may briefly be included for an idea about the problems of determining the age and providing an estimate of the time range of the megaliths in question.

All the available datings from different sites in Meghalaya and Nagaland are now added in the chapter “Megalithic monuments in Northeast India” (page 7, line 136ff.). Although these are not too many, this section now provides an understanding of the excavations and investigations already been carried out, the references as well as the immense time span which is documented for the megalithic building activities in NE-India (dates range from 270-660 cal AD to 1420-1640 cal AD). 

Page 23: Mithun (bos frontalis) should be Bos frontalis (in ittalics)

This mistake was corrected (now page 30, line 662).

Page 38: “The cooperative … allows variation, but no significant differentiation (… size of the erected stones).” This is particularly important to note that besides social complexity involved in the tradition, there is not much of difference in the size of the erected stone structure.

This important aspect was further clarified on page 41 (line 895ff.). The social complexity being present in Angami- and Chakhesang villages does influence individual abilities to erect megalithic monuments and must therefore be included both with reference to, for example, structural preconditions and the access to social networks. 

Based on the observations made in the paper, can any generalisation be made regarding the form and type of the structures applicable for megaliths found in the neighbouring areas of Northeast India?

Due to the striking variability and megalithic monuments and their embeddedness in very different forms of social organization (from more egalitarian to highly stratified societies), generalizations are not easy to make. Surely, and this information was added at the beginning of the section “Megalithic monuments in Northeast India” (page 5, line 107ff.), it can be agreed that standing stones, or menhirs, are the most common types of megalithic monuments. They are to be found in different areas, communities and among differing forms of social structures. Among these forms of standing stones, it is common that the stones are associated with either a commemoration of social events, or are memorials of the dead. The second main type, the cist burials, actually play very different roles and function in different ways; here the mentioned variability is clearly visible. 

In accordance with the theoretical outline of this paper, it further becomes apparent that the erection of megalithic monuments among Naga communities as a specific course of action, is deeply rooted within structural components of the communities involved (e.g. social networks, kin structures, political structures) and further shaped by individual actions, abilities and choices. Although similar situations could or can also be assumed for other examples of recent megalith building activities in NE-India, it would take a lot of in-depth descriptions to properly work out any kind of broader or generalized statements than the one stated above. 

Is there any other archaeological material like pottery, stone and metal artifacts observed in the surveyed villages for understanding the nature of habitation and possible links with the megaliths?

The question of the association of specific items of material culture and megalithic monuments is indeed an interesting question, which was nonetheless not a direct subject of the case study presented here. Yet, some few data from test excavation in the same district in which also the case study is situated, are available. These are now incorporated within the chapter “Results” in the subchapter “Field work and methodology” (page 15; line 331ff.).

---

## [Editor Report · Decision Letter 1]

29 Jan 2021

Societies in balance: Monumentality and feasting activities among southern Naga communities, Northeast India

PONE-D-20-31416R1

Dear Dr. Wunderlich,

We’re pleased to inform you that your manuscript has been judged scientifically suitable for publication and will be formally accepted for publication once it meets all outstanding technical requirements.

Kind regards,

Peter F. Biehl, PhD

Academic Editor

PLOS ONE
---

## [Editor Report · Acceptance letter]

10 Feb 2021

PONE-D-20-31416R1 

Societies in balance: Monumentality and feasting activities among southern Naga communities, Northeast India 

Dear Dr. Wunderlich:

I'm pleased to inform you that your manuscript has been deemed suitable for publication in PLOS ONE. Congratulations! Your manuscript is now with our production department. 

Kind regards, 

on behalf of

Dr. Peter F. Biehl 

Academic Editor

PLOS ONE